# PROMPTa—A multifaceted assessment approach to develop writing skills

**Vezolu Puro[1], Noel Anurag Prashanth Nittala[1]\*, Hariharasudan A[2]**

1 Department of Sciences, Indian Institute of Information Technology Design and Manufacturing, Kurnool, Andhra Pradesh, India, 2 Department of Language, Culture and Society, College of Engineering and Technology, SRM Institute of Science and Technology, Kattankulathur, Tamil Nadu, India

\* noel@iiitk.ac.in

## Abstract

Assessment plays a pivotal role in developing writing skills, yet traditional teacher-led evaluations often fall short in addressing learner's diverse needs, creating a substantial gap in effective writing methodologies. Lack of evidence-based studies to measure the effectiveness of assistive assessment technologies in a traditional setting, furthers the need for the study. The main objective of this study is to examine the effectiveness of PROMPTa (Performance Review of Manuscript Proficiency Tool for Assessment), a tool-based, ICT-integrated, multifaceted assessment framework that enables alternative assessment through peer, mentor and expert assessment and feedback, on improving learners' writing performance. The study employed a mixed-methods approach, involving 143 first-year undergraduates, with 92 participants forming the final sample. Students were randomly assigned to an experimental group using PROMPTa for assessment, and a control group which followed traditional classroom evaluation. The experimental group demonstrated significant improvement, with higher post-test scores compared to the control group, resulting in a noticeable difference in the performance. These findings suggest that multifaceted peer-assisted ICT-based assessment tools like PROMPTa may significantly improve writing instruction strategies in higher education. This study not only adds to the existing literature and provides supporting evidence to technology-enhanced assessment approaches but also offers practical implications for educators who seek to improve the written performance of their students employing innovative methods of evaluation.

## Introduction

Effective communication is built on a foundation of essential skills, with writing [1] being one of the most crucial skills for academic success and professional development. It enables individuals to construct and convey complex ideas, facilitating social interaction through coherent communication [2]. Writing, the most cognitively

**Data availability statement:** All relevant data are within the manuscript and its Supporting information files.

**Funding:** The author(s) received no specific funding for this work.

**Competing interests:** The authors have declared that no competing interests exist.

**Abbreviations:** AECC, Ability Enhancement Compulsory Course; CBCS, Choice-Based Credit System; FYUGP, Four-Year Undergraduate Program; ICT, Information and Communication Technology; NCF, National Curriculum Framework; NEP, National Education Policy; PROMPTa, Performance Review of Manuscript Proficiency Tool for Assessment; SCT, Social Cognitive Theory; SPSS, Statistical Package for the Social Sciences

demanding of the four language skills, requires mastery of grammar, vocabulary, and the ability to communicate clearly for diverse audiences [3,4]. Writing also involves the integration of intellectual abilities, cognitive strategies, language rules, and motivation [5,6], making it a multidimensional skill where learners must adhere to language rules, structure information logically, and transform knowledge into clear, meaningful communication [7]. Given these cognitive demands, writing serves as a critical tool for assessing intellectual development, critical thinking, and academic achievement [8]. Unlike assessments that focus solely on content recall, writing evaluates a student's ability to synthesize and organize knowledge into coherent arguments, making it an indispensable tool for measuring academic achievement and intellectual growth. Despite its importance, developing proficient writing skills remains a challenge for both students and teachers alike [3,9].

In the digital age, Information and Communication Technologies (ICT) have transformed how writing is produced, shared, and assessed. Digital platforms have extended the boundaries of writing, promoting collaboration and multimodal expression [10,11]. At the same time, the growing reliance on Artificial Intelligence (AI) has introduced concerns regarding authenticity and academic integrity [12–14]. These changes necessitate assessment practices that emphasise higher-order skills, such as originality, analytical thinking, and creativity —skills that are not easily replicated by AI. As a result, writing assessment must evolve to meet the demands of these new educational needs.

Traditional writing assessments in higher education often rely on summative evaluations, which tend to prioritize grammatical correctness and content recall over the writing process and critical engagement. These assessments may not provide learners with adequate feedback needed for improvement, nor do they promote reflective and iterative writing practices. Hence, there is a growing recognition of the need for formative assessment practices that emphasize ongoing support, feedback and revision. When alternative assessment methods are integrated through ICT platforms, such approaches can encourage interactive, learner-centered learning that promotes writing as a process rather than a product.

### Background of the study

Nagaland, a state in Northeast India, boasts of a unique education system due to its multilingual composition, characterised by its multi-tribal population, each with distinct languages and traditions. This multilingual setting poses significant challenges for education. While English has emerged as the primary medium of instruction, its effective integration within the educational framework remains a challenge. Nagamese, a local pidgin, often serves as the medium for classroom interactions, creating a gap between what is officially taught and what is spoken. Despite an emphasis on English learning, research [15] highlights that many learners still struggle with the language. One reason is the treatment of English as a subject rather than a skill [16], which leads to low proficiency in the language, particularly in writing, as students struggle to improve their writing and express their thoughts effectively.

Nagaland's education system contrasts a high literacy rate and a deeply rooted traditional approach. Described as a "one-way traffic" [17] system, education in Nagaland is teacher-centered, leaving learners as passive recipients of information "the present system of education has failed to empower learners" [15]. This approach, reflective of Gardner's "Uniform view of Schooling" [18] disregards individual learning needs and emphasizes conformity and standardized methods. This reinforces rote learning and memorization, limiting opportunities for creativity and independent thought.

A particularly concerning aspect of the current system is the approach to writing instruction, as teachers emphasize grammar, structure, and correctness over content and expression. This misplaced emphasis is especially evident at the higher secondary level [19], where students continue to rely heavily on teacher explanations and textual interpretations. Writing is often limited to taking down dictated notes, with little opportunity for independent thought or expression. Writing is often viewed primarily as a means to achieve grades, rather than as a tool for communication and creativity [20–22]. A lack of confidence in students' writing abilities leads teachers to rely on dictation [23], reinforcing passive learning and disengagement. Consequently, writing becomes disconnected from real-life contexts, with many students perceiving it as difficult and uninteresting [24].

The assessment practices in Nagaland worsen these challenges. The predominant use of traditional summative assessments, with their focus on memorization and standardized responses, fails to capture the iterative nature of the writing process, including crucial elements such as drafting, reflection, and revision [25]. This approach overlooks the development of higher-order thinking skills, such as analysis, synthesis, and evaluation, which are essential for proficient writing. Motivation, a key predictor of writing performance [26,27], is also hindered by the rigid pedagogy. However, the prevailing pedagogy, characterized by limited feedback practices and focus on error corrections rather than meaningful interaction and constructive guidance, leaves students struggling to develop the confidence and skills needed for effective writing, creating a culture of anxiety and fear around writing tasks [28].

The necessity for change aligns with national educational priorities, as outlined in both the National Curriculum Framework (NCF) 2005 and the National Education Policy (NEP) 2020. These emphasize formative assessment as a key tool for learning and skill development. The NEP 2020 advocates for a shift from summative to formative approaches, promoting continuous evaluation to support holistic learning. This study explores the transition from traditional summative assessments to technology integrated formative, learner-centered alternative assessment approaches.

## Theoretical framework

The study builds upon the unified conceptual framework that draws on Social Cognitive Theory (SCT), the Social Cognitive Model of Writing, Constructivism, and Constructionism. Together, these theories support the exploration of how alternative assessment methods, such as peer assessment and feedback, mentor assessment and feedback, and expert assessment and feedback, facilitated by ICT, enhance writing skills.

SCT, as proposed by Bandura [29] highlights the centrality of self-efficacy in influencing human motivation, emotional regulation and behaviour with its development influenced by four primary sources: (a) performance accomplishments (i.e., successful accomplishments raise mastery expectations while repeated failures lower these), (b) vicarious experience (i.e., observing identifiable models who succeed or fail), (c) verbal persuasion (i.e., being encouraged to believe in oneself), and (d) emotional arousal (i.e., relating stressful situations to personal performance). This theory emphasises the significance of social and environmental factors in shaping an individual's confidence in their ability to perform tasks. This is particularly relevant to writing, where learners' beliefs in their ability are shaped by iterative feedback and guided practice. Building on SCT, Zimmerman and Risemberg [30] recognize writing as a dynamic, self-regulated process that integrates personal, behavioural, and environmental strategies. This is done through multiple rounds of feedback. Upon receiving feedback, writers review their work and refine it through necessary changes. Cognitive writing strategies, such as structuring ideas, revising work, which are essential for improving writing skills, can be strengthened through explicit instruction and learning from peers [29,30].

Supporting the above, Constructivism [31,32] explains that learning occurs most effectively when new knowledge is connected to past experiences in a meaningful way. Writing, particularly in peer feedback, becomes a space where knowledge is actively negotiated, enabling learners to review each other's work and exchange ideas, refine their understanding through interaction and reflection, and develop deeper insights through different perspectives. Constructionism [33] builds on this idea by emphasising "learning by making," where learners engage in hands-on, creative activities. In this case, writing itself becomes an act of making, shaped and refined through iterative drafting, feedback and revisions.

The integration of these theories forms the central premise of the study, which posits that writing proficiency develops most effectively when learners are actively engaged in constructing meaning, regulating their progress, and receiving scaffolded feedback. Technology enables this framework as ICT expands the space for meaningful engagement, allowing students to collaborate, share feedback, and monitor their learning progress. It aligns with the theoretical tenets of self-regulated learning and social interaction by facilitating reflective practice, enhancing the quality of feedback, and improving writing outcomes. Thus, this integrated framework guided the study by informing the design and development of PROMPTa, the inclusion of peer, mentor, and expert assessment and feedback, and the focus on process-based writing assessment. Theoretical foundations support the hypothesis that ICT-mediated assessment and feedback loops, grounded in self-efficacy, collaborative meaning-making, and learner agency, can significantly enhance writing development in undergraduate learners.

## Literature review: Alternative assessment in improving writing skills through ICT`

To improve writing skills, assessment should not only focus on the final product but also the writing process and progress. In a language classroom, assessment should help learners demonstrate what they have learned rather than what they have not learned. In this vein, alternative assessment focuses on "assessment for learning," which helps learners understand how they learn and apply their knowledge [34].

Alternative assessment encourages active participation, reflection, and continuous improvement, making it particularly effective in teaching writing. Approaches such as peer, mentor, and expert feedback facilitate an interactive process where learners can revise and refine their work based on constructive input. Some effective methods include self-assessment [35,36], portfolio assessments [37,38], and teacher-student conferences [39,40].

Research shows that peer assessment is highly effective for improving writing skills as it allows learners to collaborate in planning, revising, and composing texts, making learning more interactive and reflective [41]. It also promotes a process approach to writing, as it facilitates negotiation, decision-making, communication, and critical thinking [42–47]. A study by Tinh [3] found that learners who regularly used peer review checklists significantly improved their writing skills. However, Peer assessment has its challenges. Studies suggest that its effectiveness depends on factors such as the number of peers involved, the quality of feedback, and trust among the peers [48,49]. There are also concerns about bias and reliability [50–52].

The introduction of ICT has expanded peer assessment, making it more flexible and accessible. Online peer feedback has demonstrated improvements in learning outcomes [53–57]. ICT-supported peer feedback offers a supportive learning environment, encouraging self-regulation and self-direction [58,59]. It also enhances critical thinking [60] and increases peer commentary, leading to more effective revisions [61]. One of the primary reasons for its success is its motivational impact, which reduces learner pressure and enables freer expression [62,63]. Furthermore, ICT minimizes time and space limitations, facilitating peer feedback [64]. Despite these advantages, ICT-facilitated peer feedback presents challenges. Anonymity can sometimes hinder the specificity of feedback [64]. Learners may also struggle with trust in peer evaluations, resistance to critical feedback, and low feedback literacy [65–67]. Providing constructive peer feedback requires higher-order thinking skills, which some students may lack [68,69].

 

A structured peer assessment process enhances its effectiveness. Clear assessment criteria help students understand what to focus on when reviewing peers' work. Peer assessment promotes responsibility for learning, aligning with modern pedagogies that emphasize independence and active participation. It cultivates critical thinking, communication, and self-reflection skills, making learners more engaged and effective. Writing is more than an individual task; it is a social activity that involves communication between the writer and the instructor [70]. Alongside peer assessment and feedback, teacher feedback, whether oral [51,70,71] or written [72,73], as well as expert assessment and feedback [74–77], play a significant role in developing writing skills.

## Research gap

Current research has a dearth of rigorous mixed-methods studies that offer enough evidence to validate the use of technology-aided feedback and assessment framework in comparison to the traditional evaluation methods. Despite studies showing the benefits of alternative assessments and feedback, there remains a lack of comprehensive studies that examines the combined use of multiple approaches such as peer, mentor and expert assessments and feedback as an integrated approach to improving writing skills [78–80]. Lack of evidence-based studies to measure the effectiveness of assistive assessment technologies in a traditional setting, furthers the need for the study. The study seeks to highlight the existing critical gaps in prevailing assessment practices and the need to bridge these gaps by employing multifaceted ICT integrated assessment tools. The findings of the study are expected to provide practical insights for educators and institutions seeking to implement more dynamic and learner-centred writing assessment strategies in writing classrooms.

## Study objectives

1. To investigate the impact of alternative assessment methods, such as peer assessment and feedback, mentor assessment and feedback, and expert assessment and feedback, on the development of writing skills in the context of Nagaland.

2. To assess the impact of ICT integration in supporting writing skill enhancement through the implementation of alternative assessment methods.

## Null hypotheses

1. There is no significant difference in the improvement of writing skills between learners who receive alternative assessments and those who receive traditional assessment methods.

2. The use of ICT-integrated alternative assessments does not significantly improve writing outcomes compared to traditional approaches without ICT.

## Methodology

The study employed convergent parallel mixed-methods research design to provide a comprehensive understanding of the research problem by combining both quantitative findings and qualitative insights, ensuring triangulation of data, thereby strengthening the validity of the statistical findings of the study in terms of measurable outcomes. The study adopted a quasi-experimental design featuring a control and an experimental group, to investigate the effectiveness of alternative assessment methods in developing writing skills. Quantitative and qualitative data were collected simultaneously, analysed independently, and then interpreted together to provide a comprehensive understanding of the intervention's impact, laying foundation for drawing conclusions.

## Research design

The quasi-experimental design compared outcomes between the two groups. The experimental group used PROMPTa, an ICT-based platform that supports multifaceted formative assessment (peer, mentor, and expert feedback), while the control group followed traditional assessment methods, where the assessment was conducted solely by the teacher. Both groups completed the same writing tasks, but their assessment processes differed significantly. Pre-test and post-test were used to quantitatively measure writing improvements. Simultaneously, qualitative data were gathered through questionnaires, semi-structured interviews, and classroom observations to explore the perceptions and experiences of the assessment practices.

## Participants of the study

The study involved first-year undergraduate learners (aged 18–22 years, mixed gender) enrolled in Ability Enhancement Compulsory Course (AECC) at a college in Kohima, Nagaland. A total of 143 first-year undergraduate students were initially selected. All participants completed a written composition pre-test to establish a baseline level of writing proficiency. To create a more homogeneous sample for the study, 92 students whose scores fell within one standard deviation of the mean were selected and randomly assigned to experimental (online intervention) and control (Offline, traditional instruction) groups of equal size (n = 46), ensuring baseline equivalence between two groups. This approach ensures adequate statistical power by reducing confounding variables that may otherwise vary, making findings more reliable and accurate despite pre-existing disparities. All participants were informed about their involvement, and written consent was obtained. The study received ethical approval from the Institutional Ethics Committee of the host institution.

## Instruments

To ensure data triangulation, the study employed a combination of questionnaires, interviews, writing tasks, and a web application called PROMPTa, which was developed specifically for the study. Questionnaires and semi-structured interviews were conducted to gather perspectives on teaching instructions and assessment. Classroom observations provided insights into the existing classroom practices. In line with the study's focus, writing tasks were employed to understand prevailing assessment practices, facilitate comparisons between traditional teaching and assessment practices and the proposed alternative assessment approach through peer, mentor, and expert feedback, and to compare traditional and ICT-based teaching methods.

**Assessment instruments and validation.** The study used written composition tasks as both pre-test and post-test instruments to measure learners' writing proficiency. Apart from this, writing tasks encompassed a range of writing styles, including communicative, creative, and academic writing, as well as prompts and supplementary materials for reference and content building. Analytical rubrics developed with input from language experts were used for evaluation, while feedback was used to gather insights. The rubrics covered four main areas: Content, Structure and Organization, Language and Vocabulary, and Overall Impact (each scored out of 10 marks, totalling 40 marks). These criteria were designed to capture both the technical and expressive dimensions of writing. Reliability was established through intra-rater and inter-rater checks using Pearson's correlation coefficient. To ensure fairness and consistency in scoring, PROMPTa followed several accuracy checks. In this context, there were three levels of assessors: peers, mentors, and experts. Peers received orientation and practice using sample writings and clear explanations of the rubrics created by the researcher. The mentor, who was also the researcher, applied the same validated rubrics when providing scoring and feedback. Experts, who had validated the rubrics, also evaluated the final drafts to ensure consistency across all stages. While scoring PROMPTa offered a structured digital platform where all assessors could refer to the same rubrics. Student identities remained anonymous to prevent bias, and the platform encouraged specific, evidence based feedback instead of subjective opinions.

Usability and accessibility were examined qualitatively through open-ended questionnaires and interviews focusing on navigation, ease of use, and accessibility of submission, feedback, and anonymity features. Though no standardized user-satisfaction scales were employed, post-intervention responses indicated strong usability: 82% found the prompts engaging and 88% appreciated the platform's accessibility. These, supported by qualitative feedback show that PROMPTa was both user-friendly and accessible.

**PROMPTa (Performance Review of Manuscript Proficiency Tool for Assessment).** PROMPTa, a web application designed specifically for the study, serves as the primary digital platform for evaluation and feedback. PROMPTa differs from traditional approaches by prioritizing the cyclical nature of writing over solely evaluating the final product. The platform acknowledges that effective writing is a developmental process that requires ongoing refinement through revisions and redrafting. This aligns with the study's objectives, which aim to explore how alternative assessment methods can promote a deeper understanding of the writing process itself. The use of PROMPTa is important for the study, as it provides a platform for alternative assessment and allows for the exploration of how integrating ICT tools can enhance writing skills.

**Instructional phases and weekly implementation plan.** The study was conducted over a duration of 8 weeks, first week included orientation, a pre-test, a questionnaire, and an introduction to writing concepts for both groups. In the second week, the experimental group was introduced to alternative assessment methods using PROMPTa, supported by brainstorming activities, while the control group followed the traditional approach. Weeks 3–7 focused on three writing tasks, namely, communicative, creative, and academic, each involving multiple drafts and iterative revisions. The experimental group received assessments guided by rubrics, as well as structured feedback from peers, mentors, and experts. Peer assessments were conducted anonymously, and the final week concluded with a post-test and a reflection session.

**Implementation of PROMPTa in the study.** In this study, PROMPTa serves as a structured submission platform for multi-tiered assessment. It relies on human-assessed feedback rather than automated feedback. Feedback is exchanged anonymously, and evaluations are conducted using analytical rubrics developed by language experts to ensure accuracy and consistency. These rubrics served as standardized guidelines, outlining clear criteria for assessing various aspects of the written drafts while allowing room for open-ended feedback to address individual strengths and areas for improvement.

The assessment process involved three writing tasks, each emphasising a distinct writing style: email writing for communicative purposes, story writing for creative expression, and essay writing for academic purposes. These writing tasks collectively represent the essential competencies required of learners in higher education and professional development. Table 1 below presents an overview of these tasks.

Each task included three different topics, each requiring learners to submit a total of nine written drafts. Each draft underwent three rounds of assessment: peer review and feedback, mentor review and feedback, and expert review and feedback. This iterative process allowed for the refinement of the written drafts.

## Data analysis

The data was analysed through quantitative and qualitative methods. Quantitative data, as shown in Fig 1, were analysed using SPSS software, beginning with pre-test data to establish baseline comparisons and confirm the validity of group

**Table 1. Overview of the tasks.**

| Tasks | Draft 1 | Draft 2 | Draft 3 |
|---|---|---|---|
| **Task 1**-Communicative Writing (Email) | Peer Assessment and Feedback | Mentor Assessment and Feedback | Expert Assessment and Feedback |
| **Task 2**- Creative Writing (Story) | Peer Assessment and Feedback | Mentor Assessment and Feedback | Expert Assessment and Feedback |
| **Task 3**-Academic Writing (Essay) | Peer Assessment and Feedback | Mentor Assessment and Feedback | Expert Assessment and Feedback |

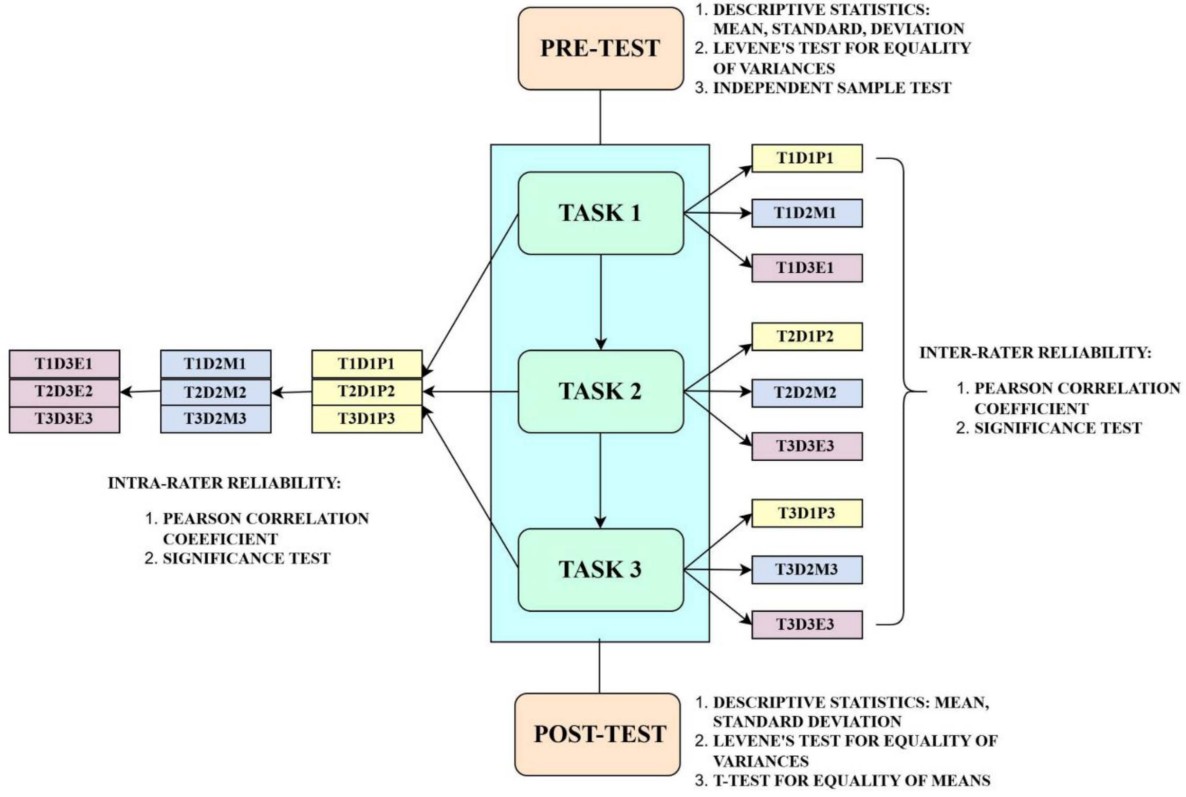

**Fig 1. Quantitative analysis.** Note: T = Tasks, D = Drafts, P = Peer-assessment, M = Mentor Assessment and Feedback, E = Expert Assessment and Feedback.

comparisons. Levene's test was conducted to check the homogeneity of variances between the groups. The reliability assessment utilized Pearson's correlation coefficient to examine both intra-rater reliability (consistency of individual ratings over time) and inter-rater reliability (agreement between different raters). Statistical significance testing was done to validate the observed correlations. To evaluate the effectiveness of the intervention, t-tests were performed on the post-test data to compare the mean differences between groups. The quantitative analysis provided measurable outcomes of the intervention's impact.

To complement the statistical analysis, a qualitative analysis was conducted through thematic analysis to identify patterns and gain insights. This dual analytical approach enabled a comprehensive understanding of both the measurable outcomes and the contextual factors influencing the intervention's effectiveness. Alongside each analysis, the discussions of the findings are also presented in this section.

## Discussion of the results

**Quantitative analysis—Results of pre-test.** The pre-test was conducted to assess the initial proficiency levels of the learners and to establish homogeneity between the Experimental and Control groups before the intervention. The analysis included statistical tests to compare the groups' performance, and the results are shown in Table 2 and Fig 2.

Levene's Test was carried out to determine if the variances between the two groups were equal. The results are as follows:

Table 2. Pre-test results for control group and experimental group.

| Group | N | Mean | SD | Levene's Test for Equality of Variances | | T | Df | Sig. |
|---|---|---|---|---|---|---|---|---|
| Experimental | 46 | 13.28 | 1.52 | 0.57 | 0.84 | 0.39 | 89 | 0.69 |
| Control | 46 | 13.41 | 1.45 | | | | | |

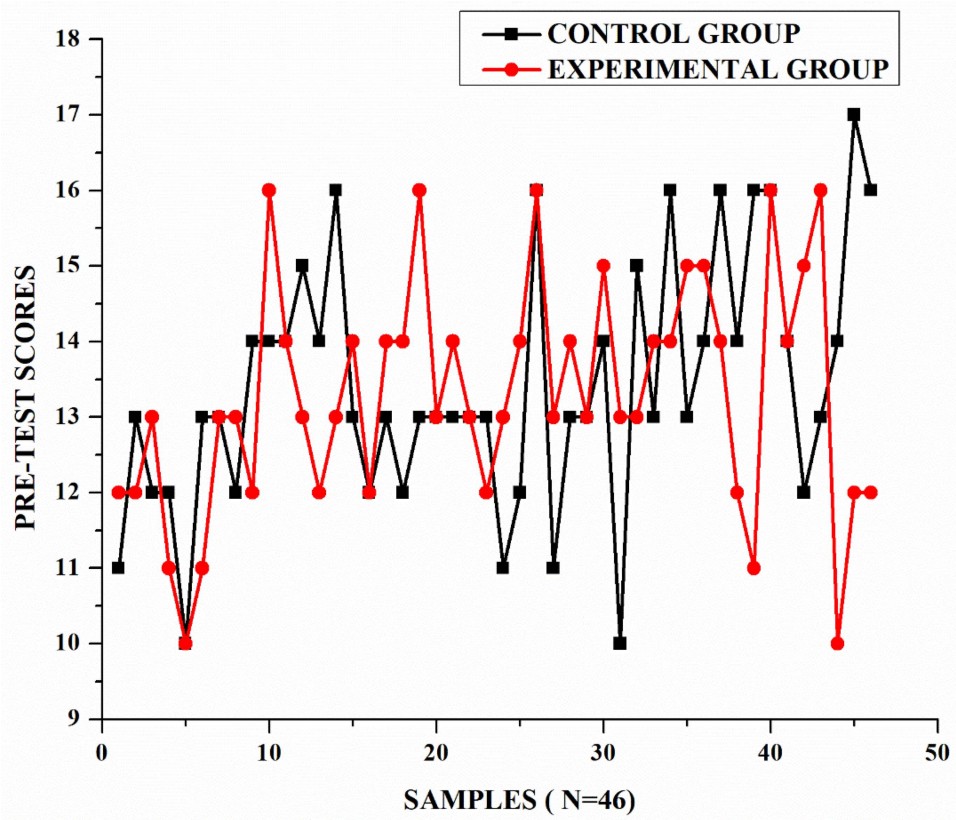

Fig 2. Graph for control group and experimental group (pre-test).

- p = 0.57 (greater than 0.05), indicating no significant difference in variances.

- This confirms that the assumption of equal variances is met.

An independent samples t-test was also conducted to determine if there was a statistically significant difference in the mean scores between the Experimental and Control groups. The results are as follows:

- t = 0.84, df = 89, p = 0.69.

- The p-value is greater than 0.05, indicating no statistically significant difference in the means.

The pre-test results demonstrate that there were no significant differences in the proficiency of the Experimental group (M = 13.28) and the Control group (M = 13.41). The equality of variances further confirms the groups' homogeneity. This indicates that the two groups were comparable in their initial proficiency levels, providing a solid baseline for evaluating the effects of the intervention. By establishing homogeneity, the analysis ensures that any differences observed in

the post-test can be attributed to the intervention rather than pre-existing differences between the groups. This baseline assessment is critical for the validity of subsequent comparisons.

**Results of reliability estimates.** To assess the consistency of post-test scores, both internal (intra) and external (inter) reliability measures were employed. This involved calculating the correlation between scores assigned by the same rater for the same tasks. The correlation between scores provided by different raters for the different tasks was also calculated.

The following sections analyze the agreement and consistency among raters involved in an evaluation process. The data is presented in two sections: Intra-Rater Reliability and Inter-Rater Reliability.

**Intra reliability estimates.** Intra-rater reliability measures the consistency of scores assigned by the same rater across multiple evaluations of the same tasks. Table 3 and Fig 3 present the Pearson correlation coefficients for the three rounds of ratings conducted by a single rater. The significance of these correlations is also reported.

The intra-rater reliability analysis revealed strong positive correlations across all three rating rounds, indicating substantial consistency in the rater's evaluations over time. The strongest correlation was observed between the first and second ratings ($r = 0.743$, $p < 0.001$), followed by the second and third ratings ($r = 0.693$, $p < 0.001$), while the first and third ratings showed a slightly lower but still strong correlation ($r = 0.667$, $p < 0.001$). All correlations were statistically significant at the $p < 0.001$ level, demonstrating the relationships were not due to chance. The gradually decreasing correlation coefficients suggest a slight drift in scoring patterns over time; however, the consistently strong correlations above 0.65 across all three rounds indicate that the rater maintained a reliable judgment throughout the evaluation process, lending credibility to the assessment outcomes. These results support the reliability of the intra-rater assessment process, ensuring the evaluator maintained a consistent standard while assigning scores.

**Inter-reliability estimates.** Inter-rater reliability measures the level of agreement and consistency among different raters evaluating the same tasks. Table 4 presents the Pearson correlation coefficients for ratings assigned by three different raters (Rater A, Rater B, and Rater C). The significance levels for the correlations are also reported.

The inter-rater reliability analysis presented in Table 4 and Fig 4 indicates a moderate to strong level of agreement among the three raters (Rater A, Rater B, and Rater C), with correlation values ranging from 0.569 to 0.752. The highest agreement was observed between Rater B and Rater C (0.752), while the lowest, though still moderate, was between Rater A and Rater C (0.569). All correlations are statistically significant at the 0.001 level, confirming the reliability of the results and ensuring that the observed agreement is not due to chance.

**Results of post-test.** The results in Table 5 and Fig 5 show a significant difference between the experimental group (M = 16.91, SD = 3.02) and the control group (M = 13.65, SD = 2.15) in post-test scores, with the experimental

**Table 3. Estimates for intra-rater reliability.**

| Ratings | Correlation | First rating | Second rating | Third rating |
|---|---|---|---|---|
| First rating | Pearson correlation | 1 | 0.743*** | 0.667*** |
| | Sig. (2-tailed) | – | 0.000 | 0.000 |
| | N | 46 | 46 | 46 |
| Second rating | Pearson correlation | 0.743*** | 1 | 0.693*** |
| | Sig. (2-tailed) | 0.000 | – | 0.000 |
| | N | 46 | 46 | 46 |
| Third rating | Pearson correlation | 0.667*** | 0.693*** | 1 |
| | Sig. (2-tailed) | 0.000 | 0.000 | – |
| | N | 46 | 46 | 46 |

**Correlation is significant at the 0.001 level (2-tailed).

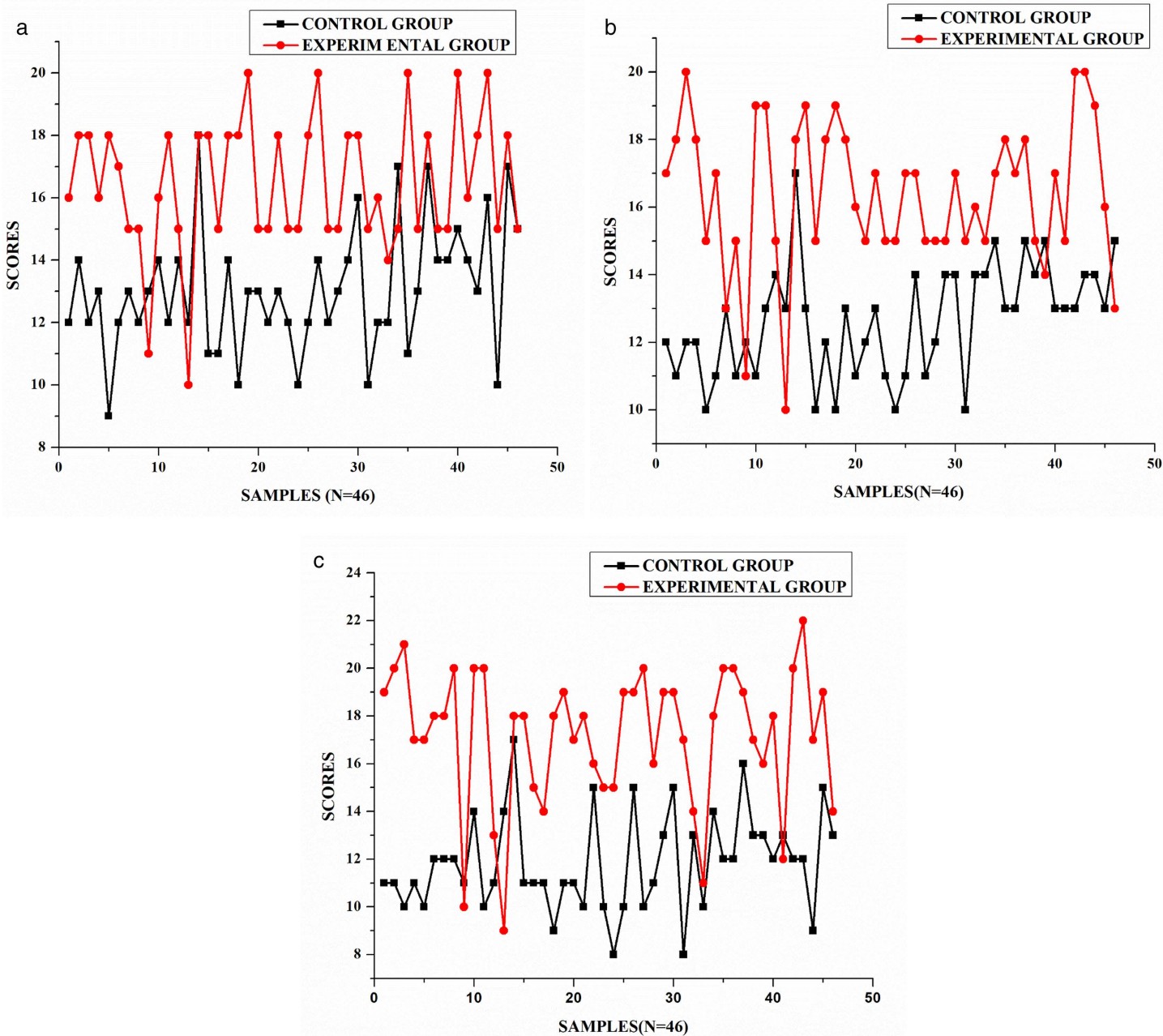

**Fig 3. Intra-rater reliability estimates for control group and experimental group.** (a) T1D2 = Control Group, T1D2M1 = Experimental Group, (b) T2D2=Control Group, T2D2M2 = Experimental Group and (c) T3D2=Control Group, T3D2M3=Experimental Group.

group outperforming the control group. The mean difference of 3.26 points was statistically significant (t (81) = 5.88, p < 0.001), indicating the intervention's effectiveness. The standardized mean difference was Cohen's d = 1.25 (95% CI [0.80, 1.69]), indicating a large effect size. This effect explains approximately 30 percent of the variance in scores, highlighting substantial practical significance beyond the p < 0.001 finding. Greater variability in the experimental group's scores (SD = 3.02) suggests differing impacts among participants. Levene's test confirmed unequal variance (p = 0.009),

**Table 4. Estimates for inter-rater reliability.**

| Raters | Correlation | Rater A | Rater B | Rater C |
|---|---|---|---|---|
| Rater A | Pearson correlation | 1 | 0.698*** | 0.569*** |
| | Sig. (2-tailed) | – | 0.000 | 0.000 |
| | N | 46 | 46 | 46 |
| Rater B | Pearson correlation | 0.698*** | 1 | 0.752*** |
| | Sig. (2-tailed) | 0.000 | – | 0.000 |
| | N | 46 | 46 | 46 |
| Rater C | Pearson correlation | 0.569*** | 0.752*** | 1 |
| | Sig. (2-tailed) | 0.000 | 0.000 | – |
| | N | 46 | 46 | 46 |

**Correlation is significant at the 0.001 level (2-tailed).

reflecting diverse response patterns. Despite this, the experimental group consistently scored higher across most of the 46 samples. The 3.26-point mean difference represents a 24 percent improvement over the control group, highlighting the intervention's practical significance.

**Comparative analysis between the groups.** This section presents a comparative analysis of pre-test and post-test scores for both control and experimental groups (N = 46 each).

As seen in Fig 6, the control group shows slight variations between pre-test and post-test scores, with scores ranging approximately from 10 to 20, indicating minimal improvement. The average difference between pre-test and post-test scores appears relatively small, suggesting that traditional assessment methods produced limited enhancement in writing skills. The experimental group, on the other hand, demonstrates a more pronounced difference between pre-test and post-test scores. While pre-test scores remain relatively consistent with the control group (ranging from approximately 10–16 points), the post-test scores show higher results, frequently reaching between 18 and 22 points. This indicates a substantial improvement in performance after implementing ICT-based alternative assessment methods.

These findings support the effectiveness of ICT-based alternative assessment in enhancing writing skills. The consistently higher post-test scores in the experimental group suggest that integrating technology into assessment can lead to more substantial improvements in student performance. Based on the analysis, it can be concluded that the intervention positively influenced the experimental group, leading to the rejection of the null hypotheses that:

1. There is no significant difference in the improvement of writing skills between learners who receive alternative assessments and those who received traditional assessment methods.

2. The use of ICT-integrated alternative assessments does not significantly improve writing outcomes compared to traditional approaches without ICT.

## Qualitative analysis

The study employed thematic analysis [81] to identify recurring themes in qualitative data from questionnaires and interviews. Krismonica et al.'s [82] framework was adopted, which grouped writing challenges into internal and external factors, to gather learners' perspectives on PROMPTa and alternative assessment. Internal challenges included grammatical difficulties, vocabulary limitations, punctuation, spelling, and content organization, while external challenges involved understanding writing stages, motivation, time constraints, practice opportunities, and feedback availability. Coding was manually done using a hybrid approach where the framework guided initial categorization while additional themes

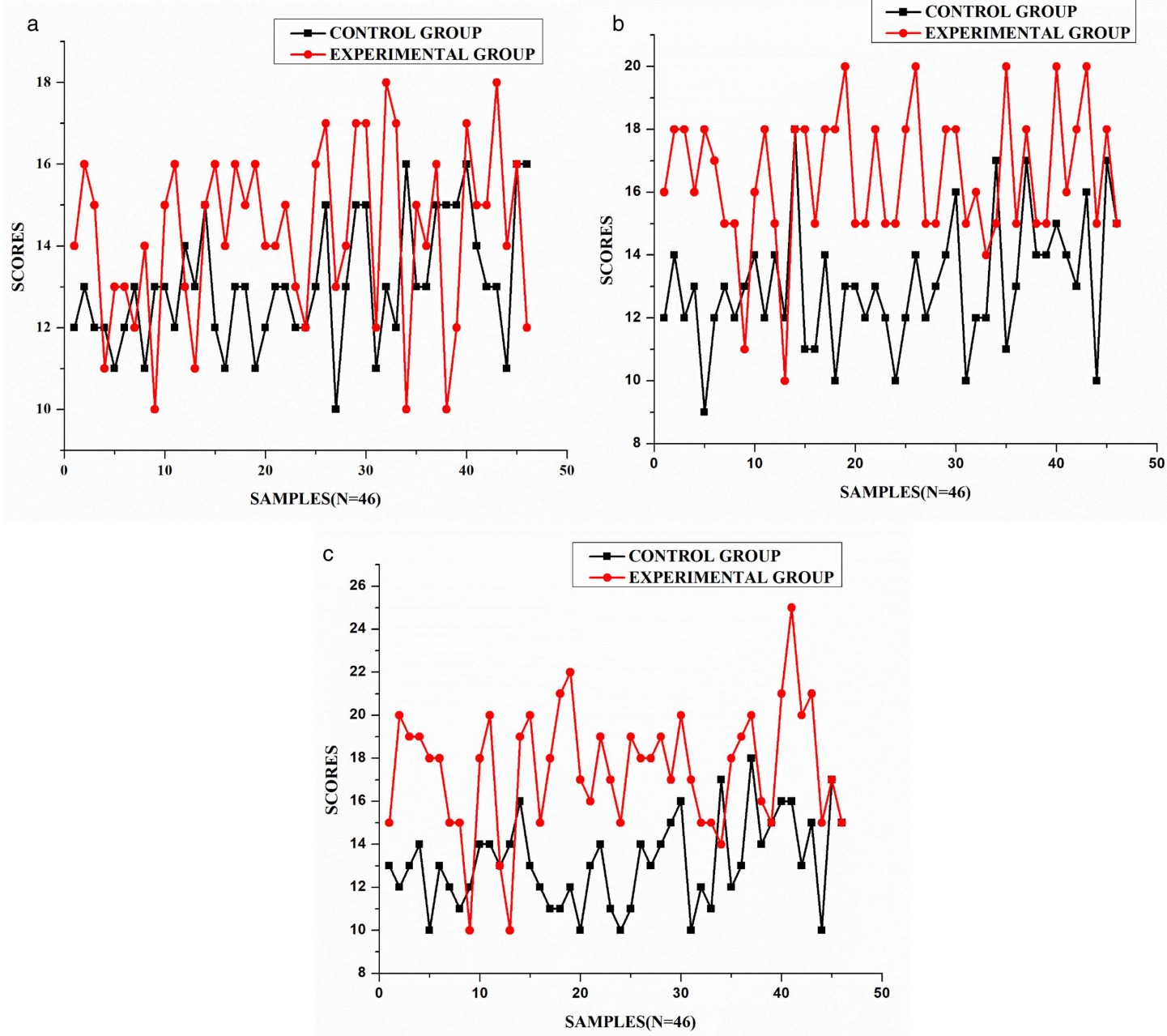

**Fig 4. Inter-reliability for both control and experimental group.** (a) T1D1=Control Group, T1D1P1=Experimental Group, (b) T1D2=Control Group, T1D2M1=Experimental Group, and (c) T1D3=Control Group, T1D3E1=Experimental Group.

emerged from participant's responses. The analysis highlights that these challenges are interconnected and can impact a learner's ability to write effectively.

Many learners struggled with grammatical issues (36 out of 46, or 78%), indicating reliance on dictated notes and rote memorization. A limited vocabulary (31 out of 46, or 68%) hindered self-expression, indicating that learners struggle to find appropriate words to convey their thoughts. Meanwhile, difficulties with punctuation and spelling (35 out of 46, or

Table 5.  Post-test results for control group and experimental group.

| Group | N | Mean | SD | Levene's Test for Equality of Variances | | t | Df | Sig. | Mean Difference |
|---|---|---|---|---|---|---|---|---|---|
| Experimental | 46 | 16.91 | 3.02 | 7.11 | 0.009 | 5.88 | 81 | 0.000 | 3.26 |
| Control | 46 | 13.65 | 2.15 | | | | | | |

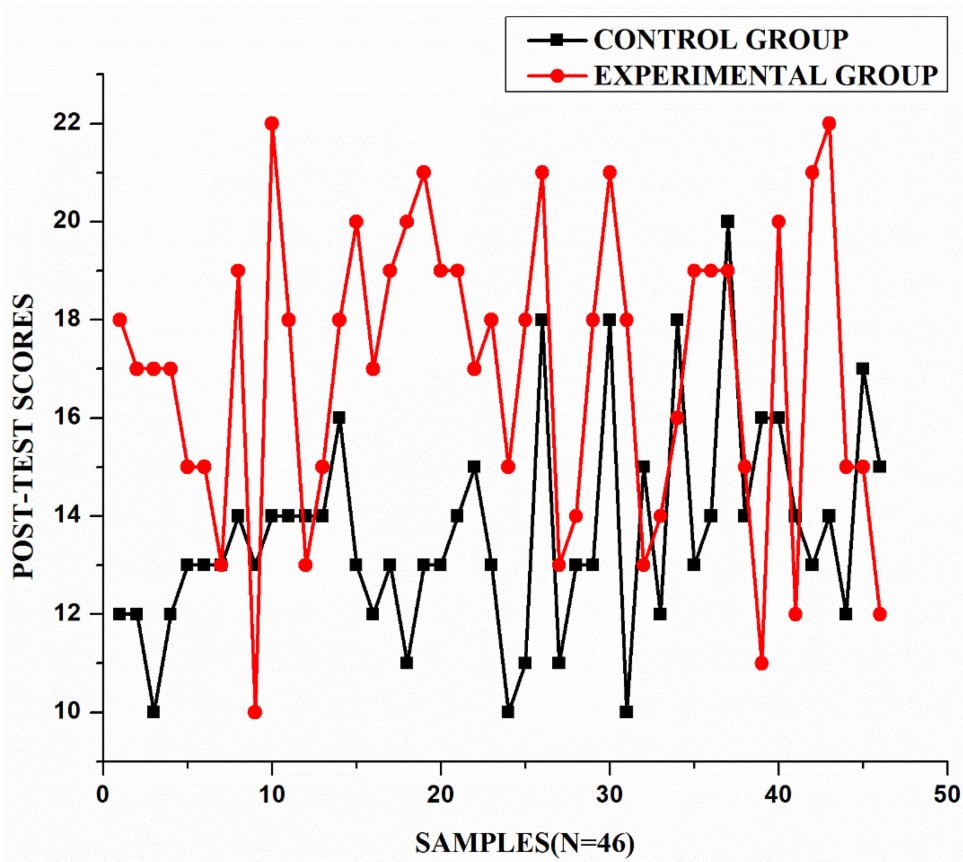

**Fig 5.  Graph for control group and experimental group (post-test).**

75%) were attributed to insufficient practice. Content organisation (38 out of 46, or 83%) was also a significant challenge, as learners lacked awareness of paragraphing and logical structuring, resulting in a disjointed writing process. External challenges were equally significant. Difficulty in understanding writing stages (43 out of 46, or 93%) indicated that learners lacked structured writing instruction, perceiving writing as a single-step task rather than an iterative process. Low motivation (41 out of 46, or 89%) stemmed from writing anxiety and a fear of criticism, with learners describing writing as a stressful and boring activity. Time constraints affected 78% (36 out of 46), as writing was viewed primarily as a means to obtain grades rather than a skill for personal development. Limited feedback was another recurring theme, with 72% (33 out of 46) reporting that teacher feedback was limited to marking errors. A strong theme was the desire for meaningful feedback, with several learners frustrated by simple error corrections. Following the intervention, contrasting themes emerged. Learners valued the structured nature of the feedback within PROMPTa, which helped them identify strengths and weaknesses, reduce anxiety, and feel more supported and engaged when revising their work collaboratively.

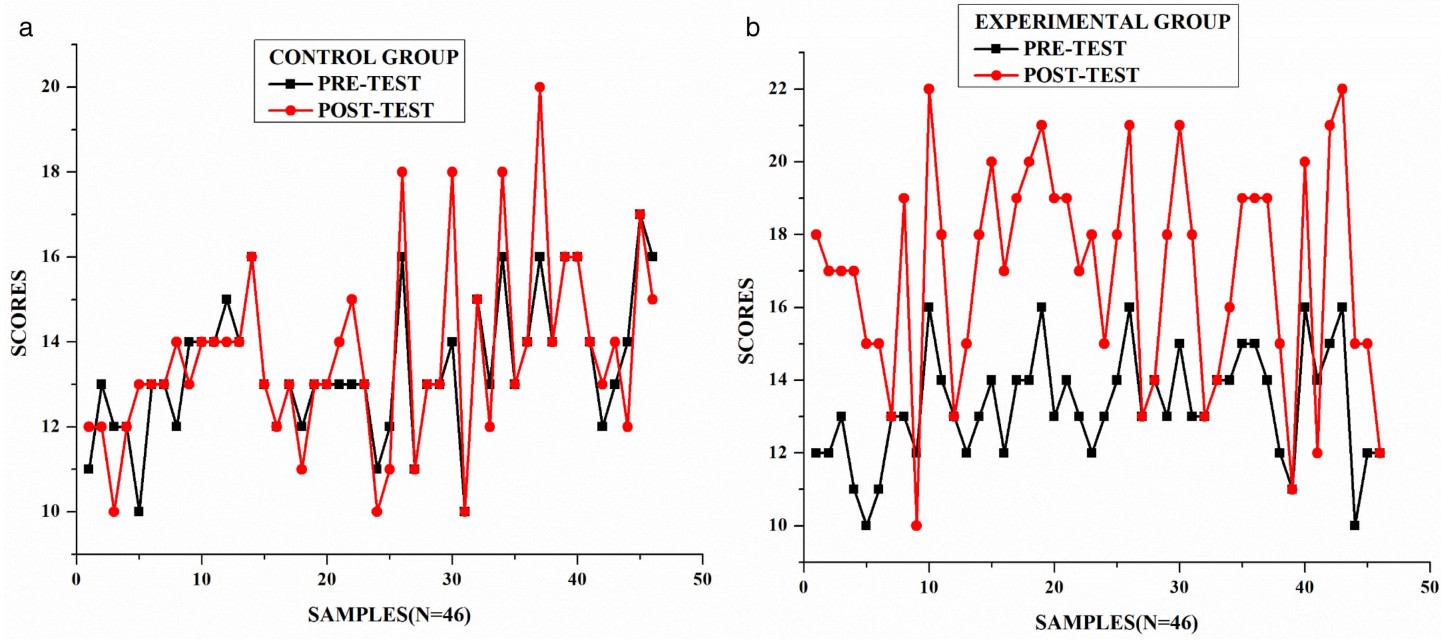

**Fig 6. Pre-test and post-test results.** (a) Control group, (b) Experimental group.

Structured guidance through PROMPTa's rubrics and prompts provided clear expectations, assessment tools and iterative feedback. This was further strengthened by the feedback learners received from mentors, peers and experts. For instance, mentor feedback focused on improving sentence clarity and structure, as seen in Fig 7.

Before the intervention, thematic analysis revealed that 86% of learners lacked confidence in writing and 92% were unaware of alternative assessment methods, as shown in Fig 8. A strong theme emerged from their lack of awareness of writing expectations, which made it difficult to evaluate their progress. Post-intervention findings revealed greater self-awareness and control over their writing process, with 78% of learners finding peer assessment and feedback very helpful, emphasising its role in encouraging reflection and revision. One learner shared, "*I really enjoyed reading comments on my writing from my peer. It was easy to understand and very useful as well*". Another strong pre-intervention theme was the lack of exposure to ICT tools, with 79% reporting they had never used such tools in classrooms. However, post-intervention responses were positive, with 82% finding the prompts engaging and 88% appreciating the structured nature of PROMPTa. Anonymity emerged as a crucial factor in boosting confidence, reducing anxiety and fear of judgement, allowing them to write more freely. One learner expressed, "*knowing that my work will be corrected by someone of my level and someone who does not know me made me feel less embarrassed. I wrote my heart out, whatever I wanted to express, I wrote it*".

A strong dissatisfaction with traditional methods was a dominant pre-intervention theme, with 94% of learners stating that feedback was rare and limited to instances when someone performed exceptionally well or poorly, often used as examples. Post-intervention, learners expressed appreciation for the feedback that helped them understand their mistakes and improve their writing. 64% of learners found peer feedback valuable as it provided insights into common writing issues. A learner stated, "*it's nice to know I am not the only one with such problems. Everyone seems to be struggling with similar problems with their writing*". Peer feedback played a crucial role in clarifying confusion and validating shared struggles, as seen in Fig 9(a)–9(c). This highlights the collaborative and empathetic tone of peer interactions, making feedback more relatable and less intimidating.

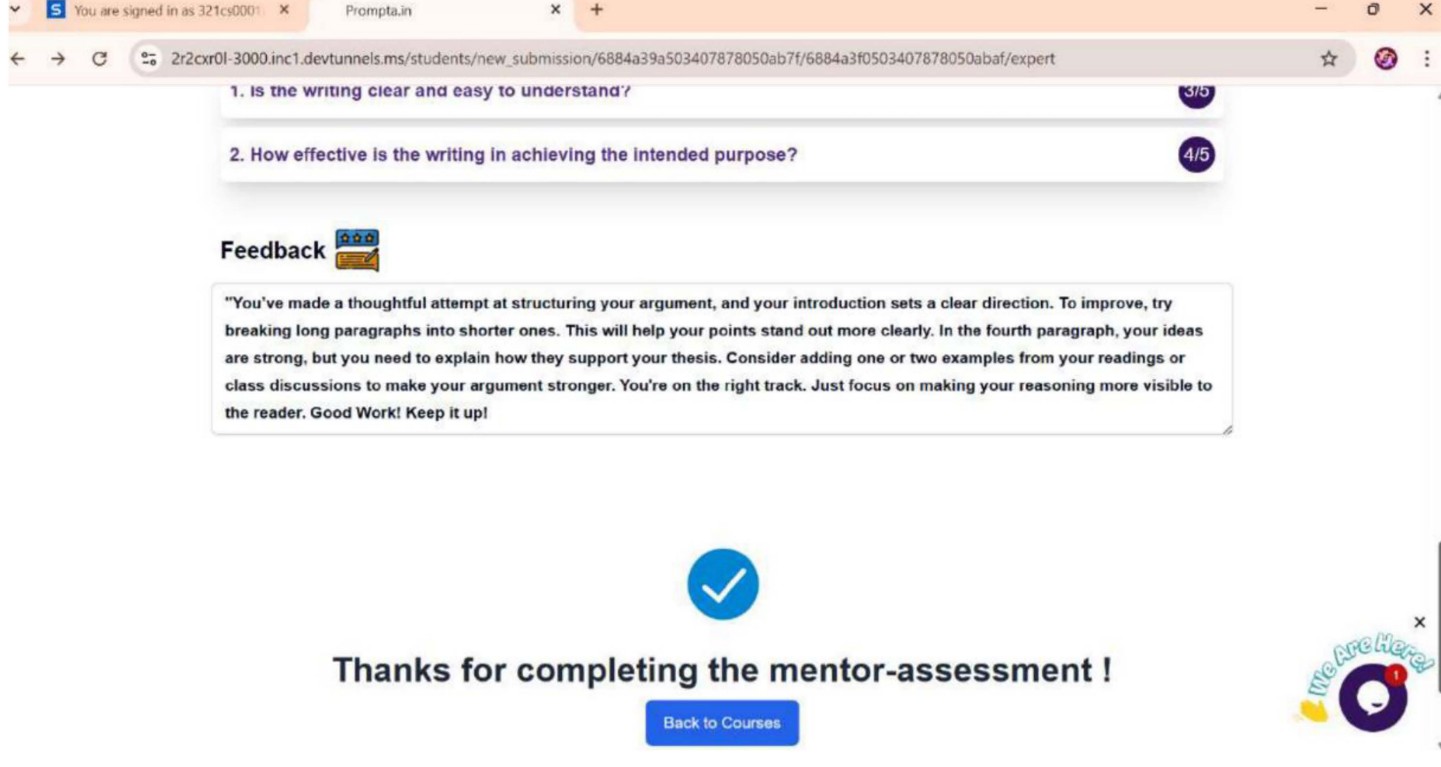

**Fig 7. Sample of mentor feedback.**

Mentor and expert feedback were also highly valued. Several learners attributed their motivation and readiness to engage in the writing tasks to the inputs gained from their feedback. Expert feedback encouraged development, as illustrated in Fig 10. Such feedback not only guided learners' revisions but also boosted their motivation and confidence.

The analysis reveals that internal and external challenges are deeply connected, primarily stemming from traditional assessment limitations and a lack of meaningful feedback. However, integration of alternative assessment through PROMPTa addressed several challenges while encouraging greater engagement and confidence, improving overall writing development.

The overall findings of this study suggest that a scaffolded approach, incorporating peer, mentor, and expert assessment and feedback via PROMPTA, contributed to improving learners' writing proficiency. The use of PROMPTa created a more interactive, learner-centered setting that encouraged self-reflection and iterative learning. Even though the outcomes are context-specific, they are aligned with the existing literature and strengthen the need for technology-mediated feedback systems in shifting away from traditional, memorisation-focused assessment practices to encourage an interactive, student-centred learning environment. The successful implementation of PROMPTA demonstrates that with structured support, ICT-driven assessments can address these gaps.

## Limitations

This study presents invaluable insights into how alternative assessments based on ICT can improve student learning, but like any other research, this study has limitations. It is important to understand these limitations to utilize the findings in an effective manner.

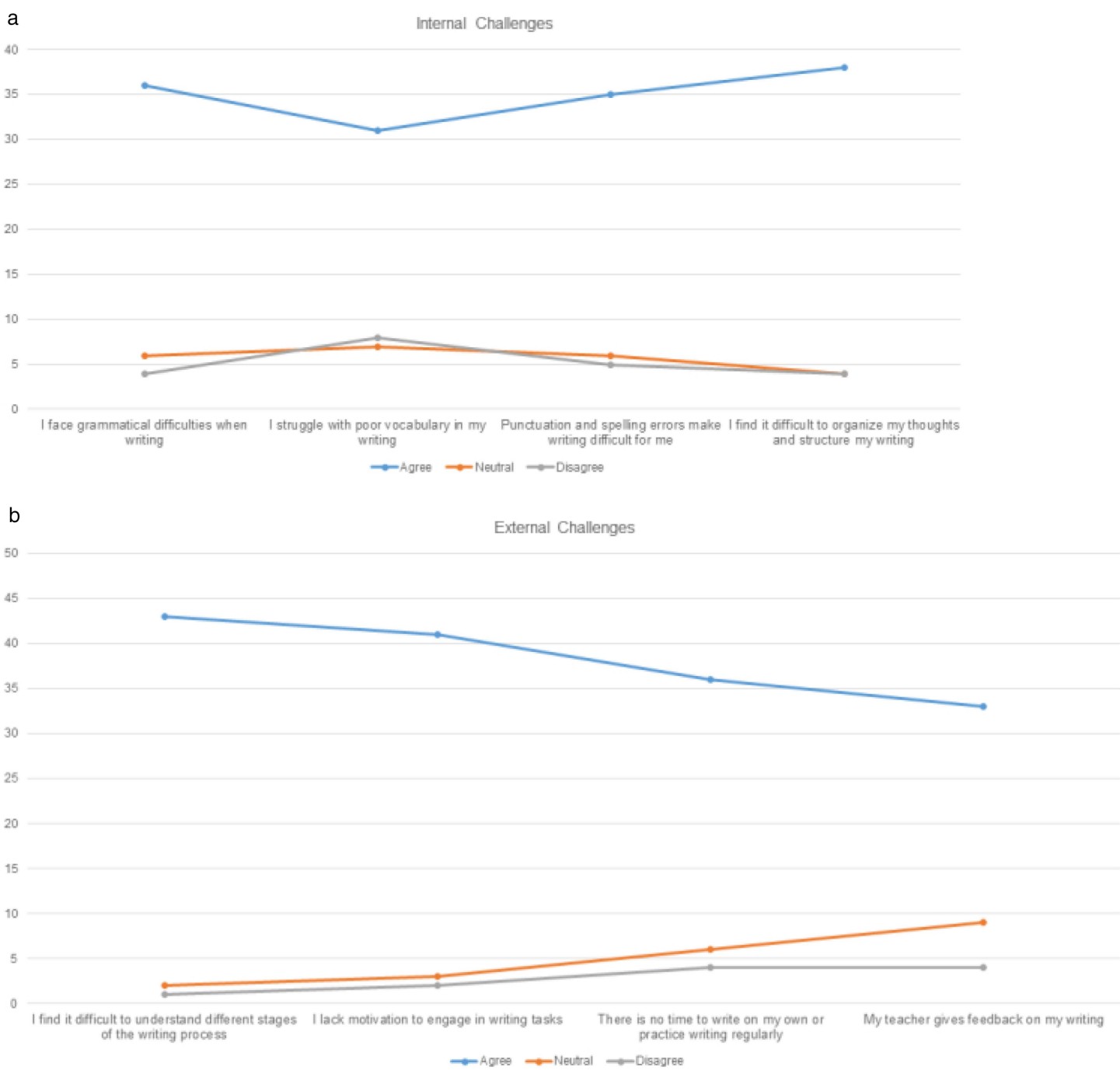

**Fig 8. Writing challenges (Krismonica et al., 2021).** (a) Internal factors (b) External factors.

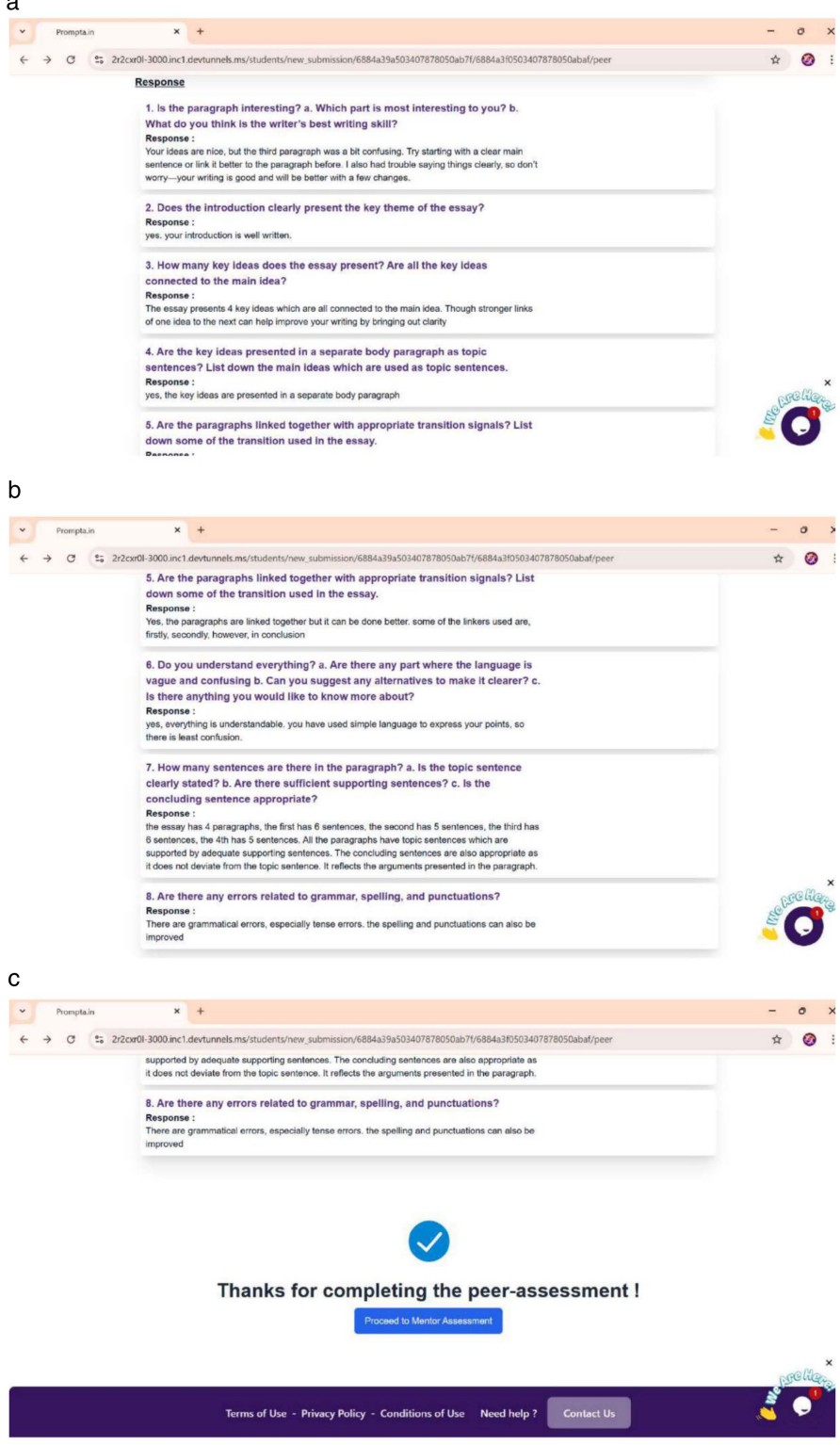

**Fig 9. Peer feedback.** (a) Sample 1 (b) Sample 2 (c) Sample 3.

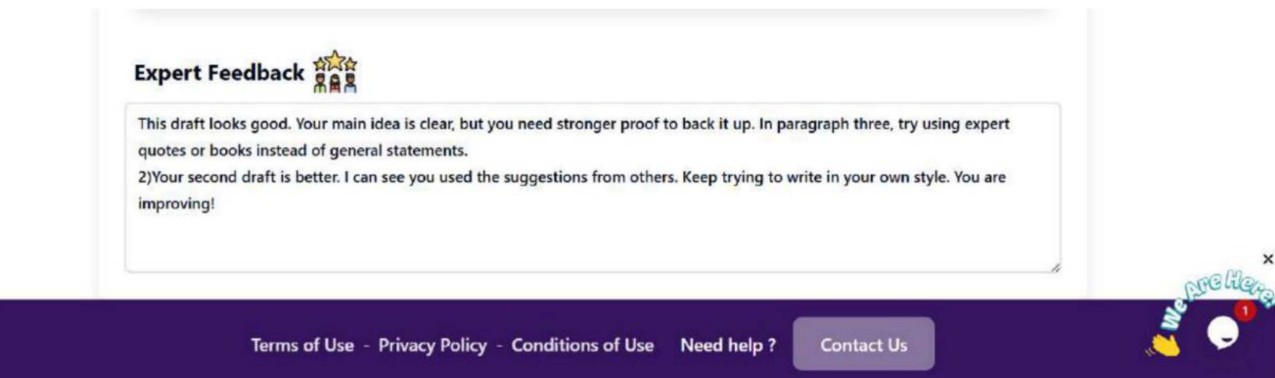

**Fig 10. Sample of expert feedback.**

1. **Sampling Limitations:** The study specifically looks only at the undergraduate students in Kohima who are undergoing AECC course. The findings of the study may not be replicable in different regions and students with different courses or at different academic levels.

2. **Duration of the Intervention:** The study was conducted over a period of 8 weeks and provided valuable insights, however a longitudinal study may offer a better understanding of long-term impact and effectiveness of PROMPTa.

3. **Infrastructure Limitations and Educator Preparedness:** Infrastructure limitations, digital literacy gaps and challenges related to accessibility and educator preparedness were evident through the study.

4. **Researcher Bias:** Although established coding frameworks like Krismonica et al.'s (2021) was employed, there is a scope for researcher's personal bias and results would be more reliable if multiple studies with different researchers were conducted.

5. **Scalability and Feasibility:** The scalability aspect of PROMPTa needs further exploration, particularly in various educational contexts as it depends on several factors such as stakeholders, infrastructure and existing educational practices and institutional constraints.

**Implications**

These limitations ultimately not only draw caution about overgeneralization but also points out the exciting potential of ICT-based alternative assessments that may transform the existing pedagogy, if implementation is carried out in the right manner. The following are the implications of the study:

1. **For Researchers:** Further studies including diverse groups of students over a longer period, employing multiple researchers to reduce bias may result in stronger evidence.

2. **For Educators:** The findings of the study are encouraging and adopting ICT-based alternative assessments in the actual classroom over a longer duration or a semester may have a positive impact on the students.

3. **For Policy Makers:** Commission studies across diverse student populations, various institutions, regions, academic levels and educational contexts need to be conducted before large scale implementation of ICT-based alternative assessments.

4. **Scalability and Adaptability:** PROMPTa demonstrates strong potential for adaptation beyond its original setting. Its rubric based, flexible design allows customization across subjects, grade levels, and institution types.

**Theoretical contributions.** This study contributes to the Social Cognitive Theory (SCT) and Constructivist learning theories by demonstrating how multifaceted feedback, when delivered through ICT, supports self-regulation, peer reviews, and learner reflection, which form the core principles of SCT. It also aligns with Constructionist perspectives by showing how students actively build writing skills through iterative creation, evaluation, and revision. By integrating these theories within a digital framework, the study provides an updated conceptual model for understanding writing development in technology-enhanced settings. The findings suggest that ICT-facilitated peer, mentor, and expert feedback can operationalize these theories in concrete and measurable ways, thus advancing their relevance in 21st-century educational practice.

## Conclusion

This study examined the critical gap of how teaching writing and traditional assessment led by teachers is often inadequate, and the potential of ICT-based alternative assessments, particularly through the PROMPTa platform, in supporting the writing development of undergraduate learners in Nagaland is explored. The findings of the study revealed compelling evidence that the robust design of PROMPTa in addition to the multifaceted feedback resulted in enhanced efficiency of students in writing. The adaptability of PROMPTA, particularly its functionality in low-connectivity environments, makes it a feasible, practical and inclusive tool for education in regions such as Nagaland, where digital access remains limited. The higher post test scores of the experimental group highlights the impact of a modern and innovative ICT integrated approach in addressing students' diverse learning needs. The results indicate a meaningful shift from traditional one-size fits all methods to and reiterate the fact that alternative assessment is practically effective.

Ultimately, this study's significance extends beyond mere test scores. Embracing innovative technologies and student centric frameworks will meaningfully shape writing pedagogy, resulting in effective assessment strategies and improved writing efficiency. Going forward, educators and researchers should constantly rethink how best can writing be taught and evaluated, and explore plausible ways to enhance writing proficiency, thereby preparing learners for academia and beyond.

## Supporting information

**S1 Data. Scores of written tasks for all samples, including pre-test results, multiple drafts for the three writing tasks, and post-test scores for both the experimental and control groups.**
(XLSX)

## Acknowledgments

The authors are thankful to the participants of the study and the management of the institution where the study was conducted.

## Author contributions

**Conceptualization:** Vezolu Puro, Noel Anurag Prashanth Nittala.

**Data curation:** Hariharasudan A.

**Formal analysis:** Vezolu Puro, Hariharasudan A.

**Investigation:** Vezolu Puro.

**Methodology:** Vezolu Puro, Noel Anurag Prashanth Nittala.

**Software:** Hariharasudan A.

**Supervision:** Noel Anurag Prashanth Nittala.

**Validation:** Vezolu Puro, Hariharasudan A.

**Visualization:** Noel Anurag Prashanth Nittala.

**Writing – original draft:** Vezolu Puro.

**Writing – review & editing:** Noel Anurag Prashanth Nittala, Hariharasudan A.

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
