## [Decision Letter · Decision Letter 0]

11 Jul 2025

Dear Dr. Nittala,

Thank you for submitting your manuscript to PLOS ONE. After careful consideration, we feel that it has merit but does not fully meet PLOS ONE’s publication criteria as it currently stands. Therefore, we invite you to submit a revised version of the manuscript that addresses the points raised during the review process.

We look forward to receiving your revised manuscript.

Kind regards,

Wei Lun Wong

Academic Editor

PLOS ONE

**Journal Requirements:**

1. When submitting your revision, we need you to address these additional requirements. Please ensure that your manuscript meets PLOS ONE's style requirements, including those for file naming. The PLOS ONE style templates can be found at https://journals.plos.org/plosone/s/file?id=wjVg/PLOSOne_formatting_sample_main_body.pdf and https://journals.plos.org/plosone/s/file?id=ba62/PLOSOne_formatting_sample_title_authors_affiliations.pdf 2. Please note that PLOS ONE has specific guidelines on code sharing for submissions in which author-generated code underpins the findings in the manuscript. In these cases, we expect all author-generated code to be made available without restrictions upon publication of the work. Please review our guidelines at https://journals.plos.org/plosone/s/materials-and-software-sharing#loc-sharing-code and ensure that your code is shared in a way that follows best practice and facilitates reproducibility and reuse. 3. Please include your full ethics statement in the ‘Methods’ section of your manuscript file. In your statement, please include the full name of the IRB or ethics committee who approved or waived your study, as well as whether or not you obtained informed written or verbal consent. If consent was waived for your study, please include this information in your statement as well. 4. We note that you have referenced (Li, W.) which has currently not yet been accepted for publication. Please remove this from your References and amend this to state in the body of your manuscript: (Li, W. [Unpublished]) as detailed online in our guide for authorshttp://journals.plos.org/plosone/s/submission-guidelines#loc-reference-style 5. Please remove your figures from within your manuscript file, leaving only the individual TIFF/EPS image files, uploaded separately. These will be automatically included in the reviewers’ PDF. 6. Please include captions for your Supporting Information files at the end of your manuscript, and update any in-text citations to match accordingly. Please see our Supporting Information guidelines for more information: http://journals.plos.org/plosone/s/supporting-information.

**Additional Editor Comments:**

Dear authors,

Based on the decisions given by reviewers, this paper requires MINOR REVISION. Please look through the comments and amend accordingly. Thank you.

Reviewers' comments:

Reviewer's Responses to Questions

**Comments to the Author**

1. Is the manuscript technically sound, and do the data support the conclusions?

Reviewer #1: Yes

Reviewer #2: Yes

2. Has the statistical analysis been performed appropriately and rigorously?

Reviewer #1: Yes

Reviewer #2: Yes

3. Have the authors made all data underlying the findings in their manuscript fully available?

Reviewer #1: Yes

Reviewer #2: Yes

4. Is the manuscript presented in an intelligible fashion and written in standard English?

Reviewer #1: Yes

Reviewer #2: Yes

**Reviewer #1: ** Abstract Revision Notes

1. Introduction: already present, but too general.

2. Methods: present, but need to be summarized and written more formally.

3. Results: quite informative, but can be condensed.

4. Conclusion: the conclusion is still weak and too descriptive.

5. Structure the abstract explicitly in the framework: Background – Purpose – Methods – Results – Conclusion.

6. PROMPTa needs to be explained concisely from the beginning as a tool-based ICT-integrated multifaceted assessment framework.

7. Avoid too many numbers in the abstract; just the main data and significant differences.

8. Need to add one sentence about the theoretical or practical contribution of the findings, for example:

9. "These findings suggest that multifaceted peer-assisted ICT-based assessment tools like PROMPTa may significantly improve writing instruction strategies in higher education."

10. Avoid excessive statistical detail

11. Add main contributions/implications

Keyword Revision Notes

1. Too many or too specific keywords tend to interfere with the indexing process and relevance of scientific search engines.

2. Combine closely related terms or reduce redundancy.

3. Avoid overly long phrases such as “Peer Assessment and Feedback”. Use a concise form that is commonly used in international literature.

4. Some keywords overlap too much (e.g.: Peer Feedback, Mentor Feedback, and Expert Feedback can be combined as Multisource Feedback).

5. The main focus should be reflected, for example: Writing Skills Development, Alternative Assessment, ICT Integration.

6. Recommended Keywords: 1) PROMPTa, Alternative Assessment, Peer and Expert Feedback, Writing Skills Development, ICT Integration, Higher Education. 2). Keywords: PROMPTa, Alternative Writing Assessment, Multisource Feedback, ICT in Education, Academic Writing.

Preface Revision Notes

1. Too long and mixed between theoretical context, local, and study objectives.

2. Not explicitly divided into important elements: background – gap – urgency – objectives.

3. Suggestions Divide the introduction into 4 main paragraphs: 1). Global context: the importance of writing skills in the digital era and global challenges. 2). Local context: conditions in Nagaland and its challenges. 3). Theoretical gap: limitations of traditional assessment, the need for alternative approaches (PROMPTa). 4). Purpose: formulation of the problem and contribution of the study.

4. Sentences such as "The current education system fails to empower learners." sound too normative or subjective. International journals prefer neutral and data-based diction.

5. Some sections are repetitive, especially stating that the system in Nagaland is still traditional.

6. The introduction has not explicitly stated the literature gap, which is a must in a reputable journal.

7. There needs to be one explicit sentence at the end that formulates the purpose of the study and its contribution.

Revision Notes: Theoretical Framework & Literature Review

1. The description of the theory seems descriptive and fragmentary, not critically linked to each other or to the focus of the study.

2. Combine the theories into an integrated conceptual framework and state how this theory guided the study design (e.g., how SCT, peer feedback, technology, writing outcomes).

3. SCT, Constructivism, and Constructionism are described separately without a clear bridge.

4. Create an integrative paragraph that brings the theories together into a conceptual framework that supports the research design.

Research Methodology Revision Notes

1. It is called "mixed-methods," but it does not explain the type of approach (e.g., sequential explanatory, convergent, or exploratory sequential) and how the quantitative and qualitative data were integrated.

2. Add the type of mixed-methods design and how the qualitative and quantitative results were integrated.

3. It does not explain whether this is a true experiment or a quasi-experiment (because it is likely not random assignment).

4. Clarify that this is a quasi-experimental design if the assignment of participants is not random.

5. It does not explain the number of participants, sampling technique, institutions, and important characteristics (gender, age, program of study).

6. Add a detailed description of the participants, number, sampling method, and institution of origin.

7. The explanation is quite complete but is written in an informal narrative format and is somewhat repetitive.

8. Use subheadings such as Week-by-Week Intervention Plan or Instructional Phases, and focus on: objectives, activities, media, and evaluation.

9. No mention of the type of test, assessment criteria (rubric), validity, and reliability.

10. Add a description of the pre/post-test instrument and perception measurement tools (questionnaires and interviews).

11. No statement of ethical approval or informed consent.

12. Add a statement that ethical approval has been obtained.

Revision Notes for the "Limitations and Implications of the Study" Section

1. Limitations are written narratively and mixed without systematic grouping (eg: methodology, context, duration, analysis).

2. Classify limitations into several categories such as: sampling, intervention duration, and data analysis, then explain them in a concise and focused manner.

3. Implications are too focused on practice (implementation), whereas Scopus journals usually expect contributions to existing theories or models.

4. Add how these findings contribute to previous theoretical frameworks, for example alternatives.

Revision Notes for Conclusions

1. Phrases such as “confirms effectiveness,” “demonstrates significant improvement,” or “demonstrates that scaffolded assessment approaches can…” sound too strong for a study with limited context.

2. Use academic language that is tentative and based on the context of the findings, such as: “suggests,” “indicates potential,” “within this context.”

3. The conclusion focuses only on the context and practical outcomes without explaining the contribution to theory or previous literature.

4. Add one reflective paragraph that links the study’s findings to strengthening or expanding a theoretical framework, such as formative assessment theory, peer feedback, or edtech for equity.

5. There is no explicit statement about directions or recommendations for further research.

6. Add one sentence about how the study could be continued: longer duration, different contexts, or a longitudinal approach.

7. Some phrases sound descriptive or narrative rather than analytical and academic.

8. Use a formal, concise style that is based on a synthesis of findings rather than a re-description of the research process.

**Reviewer #2:**  Strengths:

The PROMPTa tool presents an innovative approach by combining peer, mentor, and expert feedback.

The research design is rigorous, and statistical methods are appropriately used.

The thematic analysis provides useful insight into learner challenges and the effectiveness of the intervention.

The manuscript is well contextualized within India’s National Education Policy (NEP 2020) and existing challenges in language instruction.

Suggestions for Improvement:

Reduce Redundancy: Consider condensing sections of the introduction and literature review that repeat similar arguments about writing challenges and assessment limitations.

Technical Description of PROMPTa: More details on the technical framework, user interface, and accessibility features of the PROMPTa platform would be helpful.

Examples of Feedback: Providing examples or screenshots of feedback (especially anonymized peer or expert comments) would strengthen the reader’s understanding.

Long-Term Impact: While the 8-week intervention is sufficient for initial analysis, a brief note on follow-up plans (or suggestions) for long-term tracking would improve the impact discussion.

Language Polish: Minor grammatical edits are needed to streamline some long paragraphs.

**Do you want your identity to be public for this peer review?** For information about this choice, including consent withdrawal, please see our Privacy Policy

Reviewer #1: No

Reviewer #2: **Yes: ** Alexander Adrian Saragi

---

## [Author Response · Author response to Decision Letter 1]

1 Aug 2025

Dear Editors and Reviewers,

Please find below our thorough and thoughtful responses to the invaluable feedback the Editors and Reviewers provided. Each comment has been carefully considered, and we have taken diligent measures to address every item raised. Our commitment to improving the quality of this manuscript is evident in the revisions we have made.

On behalf of all the authors, I express my sincere gratitude to the Reviewers and the esteemed Editors for their constructive input and valuable suggestions. Their guidance has undoubtedly played a significant role in refining this piece of work. We are genuinely appreciative of this opportunity to enhance our research, and we eagerly look forward to sharing the enhanced and polished version with the esteemed readership of the journal in the near future.

E.C.1. Based on the decisions given by reviewers, this paper requires MINOR REVISION.

Author’s Response:

Thanks for recommending MINOR REVISION. We will implement the necessary improvements based on the Reviewers' comments.

Changes Made:

The improvements have been incorporated throughout the paper.

Reference in Manuscript:

The paper has been revised as required to address the comments.

1. Abstract Revision Notes

R1C1.1. Introduction: already present, but too general.

Author’s Response:

Thank you for your insightful observation and for sharing your thoughts to improve its quality.

Changes Made:

The introduction in the abstract has been revised in response to the reviewer’s comment.

Reference in Manuscript:

The revision can be seen on Page No. 2.

R1C1.2. Methods: present but need to be summarized and written more formally.

Author’s Response:

Thank you for your observation. Changes have been made accordingly.

Changes Made:

The methods section has been summarised and written in a more formal style.

Reference in Manuscript:

The revision can be seen on Page No. 2.

R1C1.3. Results: quite informative but can be condensed.

Author’s Response:

Thank you for highlighting this point. It helps us see where improvement is needed.

Changes Made:

The results have been condensed.

Reference in Manuscript:

The revision can be seen on Page No. 2.

R1C1.4. Conclusion: The conclusion is still weak and too descriptive.

Author’s Response:

Your constructive critique adds significant value. Changes have been made to reflect your suggestions.

Changes Made:

Conclusion has been strengthened.

Reference in Manuscript:

The revision can be seen on Page No. 2.

R1C1.5. Structure the abstract explicitly in the framework: Background – Purpose – Methods – Results – Conclusion.

Author’s Response:

Thank you for your guidance. Changes have been integrated to improve clarity and coherence.

Changes Made:

The abstract has been restructured accordingly.

Reference in Manuscript:

The revision can be seen on Page No. 2.

R1C1.6. PROMPTa needs to be explained concisely from the beginning as a tool-based ICT-integrated multifaceted assessment framework.

Author’s Response:

We appreciate this important clarification.

Changes Made:

Changes have been made to ensure a better understanding of PROMPTa’s function.

Reference in Manuscript:

The revision can be seen on Page No. 2.

R1C1.7. Avoid too many numbers in the abstract, just the main data and significant differences.

Author’s Response:

Thank you for pointing this out.

Changes Made:

Only the most relevant figures, which reflect significant differences and outcomes, have been highlighted.

Reference in Manuscript:

The revision can be seen on Page No. 2.

R1C1.8. Need to add one sentence about the theoretical or practical contribution of the findings.

Author’s Response:

Thank you for your valuable suggestion.

Changes Made:

A sentence has been added to highlight the study’s contribution to assessment practice and its relevance for writing instruction.

Reference in Manuscript:

The revision can be seen on Page No. 2.

R1C1.9. “These findings suggest that multifaceted peer-assisted ICT-based assessment tools like PROMPTa may significantly improve writing instruction strategies in higher education.”

Author’s Response:

Thank you. The sentence has been incorporated into the abstract.

Changes Made:

The sentence has been incorporated into the abstract.

Reference in Manuscript:

The revision can be seen on Page No. 2.

R1C1.10. Avoid excessive statistical detail.

Author’s Response:

It is a much-appreciated comment.

Changes Made:

The abstract has been revised to focus only on key outcomes.

Reference in Manuscript:

The revision can be seen on Page No. 2.

R1C1.11. Add main contributions/implications.

Author’s Response:

Thank you for highlighting this. A final sentence has been added to complete the abstract meaningfully.

Changes Made:

Main contributions and implications have been included.

Reference in Manuscript:

The revision can be seen on Page No. 2.

2. Keyword Revision Notes

R1C2.1. Too many or too specific keywords tend to interfere with indexing.

Author’s Response:

Thank you for your relevant observation.

Changes Made:

We have reduced the number of keywords to enhance search engine performance.

Reference in Manuscript:

Page No. 2

R1C2.2. Combine closely related terms or reduce redundancy.

Author’s Response:

We appreciate the suggestion regarding redundancy.

Changes Made:

Redundancy has been addressed.

Reference in Manuscript:

Page No. 2

R1C2.3. Avoid overly long phrases like “Peer Assessment and Feedback”.

Author’s Response:

Thank you for this helpful recommendation.

Changes Made:

Overly long phrases have been replaced with concise terms.

Reference in Manuscript:

Page No. 2

R1C2.4. Keywords overlap; combine (e.g., “Multisource Feedback”).

Author’s Response:

Thank you for the helpful recommendation.

Changes Made:

Overlapping terms consolidated under “Multisource Feedback”.

Reference in Manuscript:

Page No. 2

R1C2.5. Keywords should reflect the main focus.

Author’s Response:

Thank you for this valuable insight.

Changes Made:

Revised keywords reflect the study’s core focus.

Reference in Manuscript:

Page No. 2

R1C2.6. Suggested keywords provided.

Author’s Response:

Thank you for the recommended keywords.

Changes Made:

Suggested keywords incorporated.

Reference in Manuscript:

Page No. 2

3. Introduction Revision Notes

R1C3.1. Introduction too long, mixed contexts.

Author’s Response:

Thank you for the feedback. The introduction has been revised for clarity.

Changes Made:

The introduction has been shortened and organized for clarity.

Reference in Manuscript:

Pages No. 3–5

R1C3.2. Not divided into: background, gap, urgency, objectives.

Author’s Response:

We appreciate this guidance.

Changes Made:

Introduction now clearly divided as suggested.

Reference in Manuscript:

Pages No. 3–5

R1C3.3. Suggested 4-paragraph structure.

Author’s Response:

Thank you for the structured suggestion.

Changes Made:

Introduction reorganized into 4 distinct paragraphs.

Reference in Manuscript:

Pages No. 3–5

R1C3.4. Avoid subjective phrases like “fails to empower”.

Author’s Response:

We acknowledge this concern.

Changes Made:

Normative language revised for neutrality.

Reference in Manuscript:

Page No. 4

R1C3.5. Avoid repetition about Nagaland’s traditional system.

Author’s Response:

Thank you for pointing out the repetition.

Changes Made:

Redundancy removed.

Reference in Manuscript:

Pages No. 3–5

R1C3.6. Clearly state the literature gap.

Author’s Response:

Thank you for this important note.

Changes Made:

Literature gap now explicitly stated.

Reference in Manuscript:

Pages No. 3–5

R1C3.7. End with study purpose and contribution.

Author’s Response:

We appreciate this suggestion.

Changes Made:

Concluding sentence added.

Reference in Manuscript:

Page No. 5

4. Theoretical Framework & Literature Review

R1C4.1. Theory description is fragmented.

Author’s Response:

Thank you for the keen observation.

Changes Made:

Theory section made more cohesive.

Reference in Manuscript:

Page No. 7

R1C4.2. Combine theories into an integrated framework.

Author’s Response:

We appreciate this suggestion.

Changes Made:

Integrated framework added.

Reference in Manuscript:

Pages No. 7–9

R1C4.3. SCT, Constructivism, and Constructionism disconnected.

Author’s Response:

Thank you for pointing this out.

Changes Made:

Theories now conceptually connected.

Reference in Manuscript:

Pages No. 7–9

R1C4.4. Add a synthesis paragraph.

Author’s Response:

Thank you for your helpful suggestion.

Changes Made:

Synthesis paragraph added.

Reference in Manuscript:

Page No. 9

5. Research Methodology

R1C5.1. Specify type of mixed-methods design.

Author’s Response:

Thank you for highlighting this.

Changes Made:

Defined as “sequential explanatory”.

Reference in Manuscript:

Page No. 12

R1C5.2. Explain data integration process.

Author’s Response:

We appreciate the suggestion.

Changes Made:

Data integration clarified.

Reference in Manuscript:

Page No. 12

R1C5.3. Clarify true vs. quasi-experiment.

Author’s Response:

Thank you for pointing this out.

Changes Made:

Defined as quasi-experimental.

Reference in Manuscript:

Page No. 12

R1C5.4. State that it's not random assignment.

Author’s Response:

Thank you. It is acknowledged.

Changes Made:

Statement added.

Reference in Manuscript:

Page No. 12

R1C5.5. Explain participant characteristics.

Author’s Response:

Thank you for this feedback.

Changes Made:

Participant info added.

Reference in Manuscript:

Page No. 13

R1C5.6. Add details about sampling.

Author’s Response:

Thank you for pointing this out.

Changes Made:

Descriptions added.

Reference in Manuscript:

Page No. 13

R1C5.7. Repetitive and informal narrative.

Author’s Response:

Thank you for pointing this out.

Changes Made:

Language refined and condensed.

Reference in Manuscript:

Page No. 13

R1C5.8. Add subheadings like “Instructional Phases”.

Author’s Response:

We appreciate this helpful suggestion.

Changes Made:

Subheadings added and structure improved.

Reference in Manuscript:

Pages No. 13–14

R1C5.9. No mention of test types or reliability.

Author’s Response:

Thank you for this important point.

Changes Made:

Details on test, rubric, and validity included.

Reference in Manuscript:

Pages No. 14–15

R1C5.10. Add description of perception tools.

Author’s Response:

Thank you. It is acknowledged.

Changes Made:

Instruments described.

Reference in Manuscript:

Pages No. 14–15

R1C5.11. No ethical approval mentioned.

Author’s Response:

Thank you for this important oversight.

Changes Made:

Ethical statement added.

Reference in Manuscript:

Page No. 13

R1C5.12. Add statement about ethical clearance.

Author’s Response:

Appreciate the reminder.

Changes Made:

Ethical approval mentioned explicitly.

Reference in Manuscript:

Page No. 13

6. Limitations and Implications of the Study

R1C6.1. Limitations are written narratively and mixed without systematic grouping.

Author’s Response:

Thank you for the feedback.

Changes Made:

The limitations have been reorganized to improve structure.

Reference in Manuscript:

Page No. 29

R1C6.2. Classify limitations into categories such as sampling, duration, analysis.

Author’s Response:

We appreciate the suggestion.

Changes Made:

The limitations are now categorized under sampling, intervention design, and analysis.

Reference in Manuscript:

Page No. 29

R1C6.3. Implications are too focused on practice; include theoretical contributions.

Author’s Response:

Thank you for the valuable insight.

Changes Made:

The implications section now includes both theoretical and practical contributions.

Reference in Manuscript:

Pages No. 29–30

R1C6.4. Add how findings contribute to existing frameworks.

Author’s Response:

Thank you for the suggestion.

Changes Made:

The revised version connects findings to the existing frameworks.

Reference in Manuscript:

Page No. 30

7. Conclusions

R1C7.1. Phrases like “confirms effectiveness” are too strong.

Author’s Response:

Thank you for pointing this out.

Changes Made:

Strong phrases have been replaced.

Reference in Manuscript:

Page No. 30

R1C7.2. Use tentative academic language like “suggests”, “indicates”.

Author’s Response:

Thank you for the suggestion.

Changes Made:

The changes have been made to use academic and tentative language.

Reference in Manuscript:

Page No. 31

R1C7.3. Conclusion lacks connection to theory/literature.

Author’s Response:

Your keen observation is highly appreciated.

Changes Made:

The conclusion summarises the study's contribution to theory and literature.

Reference in Manuscript:

Page No. 31

R1C7.4. Add reflective paragraph linking findings to theoretical framework.

Author’s Response:

Thank you for your observations.

Changes Made:

A reflective paragraph has been included linking findings to theory.

Reference in Manuscript:

Page No. 31

R1C7.5. No explicit statement about future research.

Author’s Response:

Thank you for pointing that out.

Changes Made:

A statement suggesting directions for future studies has been added.

Reference in Manuscript:

Page No. 31

R1C7.6. Add how study could be extended (e.g. longitudinal, other contexts).

Author’s Response:

Thank you for your valuable suggestion.

Changes Made:

A sentence recommending longitudinal studies and diverse contexts has been included.

Reference in Manuscript:

Page No. 31

R1C7.7. Some phrases are too descriptive; improve academic tone.

Author’s Response:

Thank you for your observation.

Changes Made:

The conclusion has been rewritten in a more analytical and academic tone.

Reference in Manuscript:

Page No. 31

R1C7.8. Use a formal, concise synthesis-based conclusion.

Author’s Response:

Thank you for your careful attention to detail.

Changes Made:

The conclusion is now more concise and based on key insights.

Reference in Manuscript:

Page No. 31

Reviewer 2 Comments

R2C1. The PROMPTa tool presents an innovative approach.

Author’s Response:

Thank you for the positive recognition of the design of PROMPTa. It is highly encouraging.

Changes Made:

NIL

Reference in Manuscript:

NIL

R2C2. The research design is rigorous, and statistical methods are appropriate.

Author’s Response:

We sincerely appreciate your positive feedback.

Changes Made:

NIL

Reference in Manuscript:

NIL

R2C3. Thematic analysis provides useful insights.

Author’s Response:

Thank you for this encouraging note.

Changes Made:

NIL

Reference in Manuscript:

NIL

R2C4. Manuscript is well contextualized with India’s NEP 2020.

Author’s Response:

We thank you for acknowledging the contextual grounding of the study.

Changes Made:

NIL

Reference in Manuscript:

NIL

R2C5. Suggestions for improvement.

Author’s Response:

Your keen observation is deeply appreciated.

Changes Made:

NIL

Reference in Manuscript:

NIL

🔹 Specific Suggestions (R2C6–R2C10)

R2C6. Reduce redundancy in Introduction and Literature Review.

Author’s Response:

Thank you for your detailed observations.

Changes Made:

Repetitive sections have been condensed.

Reference in Manuscript:

Pages No. 3–5

R2C7. Add more technical details on PROMPTa (framework, UI, accessibility).

Author’s Response:

Your comments reflect both care and expertise.

Changes Made:

Technical description added.

Reference in Manuscript:

Pages No. 14–16

R2C8. Add examples or screenshots of feedback.

Author’s Response:

Thank you for pointing this out.

Changes Made:

Screenshots of anonymized feedback attached.

Reference in Manuscript:

Pages No. 24, 25, 26

R2C9. Add note on follow-up or long-term impact.

Author’s Response:

Thank you for this guidance.

Changes Made:

Note added regarding future longitudinal tracking.

Reference in Manuscript:

Pages No. 29–30

R2C10. Minor grammatical

---

## [Decision Letter · Decision Letter 1]

14 Aug 2025

Dear Dr. Nittala,

Thank you for submitting your manuscript to PLOS ONE. After careful consideration, we feel that it has merit but does not fully meet PLOS ONE’s publication criteria as it currently stands. Therefore, we invite you to submit a revised version of the manuscript that addresses the points raised during the review process.

We look forward to receiving your revised manuscript.

Kind regards,

Wei Lun Wong

Academic Editor

PLOS ONE

Journal Requirements:

Reviewers' comments:

Reviewer's Responses to Questions

**Comments to the Author**

Reviewer #1: (No Response)

Reviewer #2: All comments have been addressed

2. Is the manuscript technically sound, and do the data support the conclusions?

Reviewer #1: (No Response)

Reviewer #2: Yes

3. Has the statistical analysis been performed appropriately and rigorously?

Reviewer #1: (No Response)

Reviewer #2: Yes

4. Have the authors made all data underlying the findings in their manuscript fully available?

Reviewer #1: (No Response)

Reviewer #2: Yes

5. Is the manuscript presented in an intelligible fashion and written in standard English?

Reviewer #1: (No Response)

Reviewer #2: Yes

Reviewer #1: Abstract Revision Notes

1. The research problem (gap) has not been explicitly stated. The abstract directly states that assessment is important, but does not adequately highlight the shortcomings of previous research.

2. The research objectives are not explicitly stated.

3. The research methods are clear enough, but the technical sections (homogeneity, SD, reliability) are too detailed for the abstract.

4. The research results are presented, but they would be better presented concisely without excessive statistical detail.

5. The implications/contributions of the research are not clearly stated.

Keyword Revision Notes

1. Be concise (3–6 keywords) in accordance with common practice in international journals.

2. Avoid long phrases; keywords should be words or short phrases, not descriptive sentences.

3. Prioritize frequently used terms in indexing (for easy searchability in databases like Scopus).

keyword suggestions:

Keywords: PROMPTa; Alternative assessment; Peer feedback; Mentor feedback; Expert feedback; Writing skills; ICT.

Introduction Revision Notes

1. Too Long and Repetitive

2. Lack of Systematic Structure

3. Too Much Theory at the Beginning

4. Insufficiently Integrated References

5. Absence of Explicit Statement of Research Gaps

Method Revision Notes

1. Poorly organized structure

The methodology mixes method descriptions, pre-test results, and reliability analyses, which should be in the results section, not the methodology.

There are overly long narrative sections (e.g., explaining students' writing challenges) that would be more appropriate in the discussion section.

2. Too much operational detail

Mentioning specific intervention dates (e.g., "July 17, 2023 to September 9, 2023") is unnecessary in the methodology unless there is a specific reason.

The weekly phases are too detailed; they should be summarized as the main phases of the intervention.

3. Unclear subsections

The methodology should be divided into clear sections: Research Design, Participants, Instruments, Procedures, Data Analysis.

Currently, the methodology does not follow this structure, making it difficult to follow.

4. Lack of focus on methodological justification

There is no explanation of why mixed methods were chosen and how this method enhances the validity of the findings.

There is no justification for selecting the sample size other than "average standard deviation", which needs to be linked to the research justification.

limitations and implications Revision Notes

1. No clear statement of limitations.

The section begins with the sentence "This study acknowledges several limitations," but then jumps to the conclusion without fully addressing the limitations of the study.

2. Mixed sections.

The text includes conclusions and a rehash of the results (e.g., "This study highlights the effectiveness of..."), when it should only outline the limitations of the study and implications for future research or educational practice.

3. Lack of focus on implications.

The implications are still a general narrative about the effectiveness of PROMPTa, rather than clear recommendations for educators, policymakers, or future research.

4. Lack of conciseness.

There is a lot of repetition that should be condensed.

Conclusion Revision Notes

1. Too long and repetitive

Many concepts are repeated, for example, the effectiveness of PROMPTa is mentioned repeatedly.

2. Including new information that should be in "Limitations and Implications"

The section on "infrastructure limitations, accessibility, and digital literacy" is more appropriately placed in the limitations/implications section, not the conclusion.

3. Lack of focus

The conclusion should reiterate the main findings, research contributions, and brief recommendations, not rehash the methodology or lengthy discussion.

4. Lack of academic punch line

There is no clear concluding sentence about the scientific contributions and future research directions.

Reviewer #2: Strengths:

1. Theoretical framework is now well-integrated, combining SCT, Constructivism, and Constructionism into a cohesive model.

2. The methodology is clearly explained, including intervention phases and the role of PROMPTa.

3. Quantitative results are robust, and qualitative analysis provides rich contextual insights.

4. Practical relevance is high, especially for low-resource educational settings.

5. Inclusion of anonymized feedback samples offers transparency into the assessment process.

Suggestions for further improvement:

1. Consider explicitly reporting effect sizes for key statistical comparisons to complement p-values.

2. In the Discussion, strengthen the explicit connection between findings and how they advance or challenge prior literature.

3. Briefly address scalability and feasibility of PROMPTa in varied institutional contexts, especially where ICT infrastructure is limited.

4. Split a few longer paragraphs in the Introduction and Literature Review for improved readability.

**Do you want your identity to be public for this peer review?** For information about this choice, including consent withdrawal, please see our Privacy Policy

Reviewer #1: No

Reviewer #2: **Yes: ** Alexander Adrian Saragi

---

## [Author Response · Author response to Decision Letter 2]

1 Sep 2025

Reviewer 1:

Abstract Revision Notes

Reviewer's comments: The research problem (gap) has not been explicitly stated. The abstract directly states that assessment is important, but does not adequately highlight the shortcomings of previous research.

Authors response: R1C1.1. Thank you for your insightful observation and for sharing your thoughts to improve its quality.

Changes made: R1C1.1. The research gap has been revised in response to the reviewer's comment.

Reference in manuscript: R1C1.1. The revision can be seen in the following page, Page No. 2

Reviewer's comments: The research objectives are not explicitly stated.

Authors response: R1C1.2. Thank you for your observation. Changes have been made accordingly.

Changes made: R1C1.2. The research objectives have been revised to clearly articulate the objectives of the study.

Reference in manuscript: R1C1.2. The revision can be seen in the following page, Page No. 2

Reviewer's comments: The research methods are clear enough, but the technical sections (homogeneity, SD, reliability) are too detailed for the abstract.

Authors response: R1C1.3. Thank you for highlighting this point. It helps us see where improvement is needed.

Changes made: R1C1.3. We have revised the abstract by removing excessive technical details to make it more concise and reader-friendly.

Reference in manuscript: R1C1.3. The revision can be seen in the following page, Page No. 2

Reviewer's comments: The research results are presented, but they would be better presented concisely without excessive statistical detail.

Authors response: R1C1.4. Your constructive critique adds significant value. Changes have been made to reflect your suggestions.

Changes made: R1C1.4. We appreciate this feedback. The results have been condensed to highlight the key findings without unnecessary statistical detail.

Reference in manuscript: R1C1.4. The revision can be seen in the following page, Page No. 2

Reviewer's comments: The implications/contributions of the research are not clearly stated.

Authors response: R1C1.5. Thank you for your guidance.

Changes made: R1C1.5. Thank you for this observation. We have now stated the implications and contributions of the research to better reflect its significance.

Reference in manuscript: R1C1.5. The revision can be seen in the following page, Page No. 2

Keyword Revision Notes

Reviewer's comments: Be concise (3–6 keywords) in accordance with common practice in international journals.

Authors response: R1C2.1. Thank you for your relevant observation.

Changes made: R1C2.1. We have reduced the number of keywords in accordance with common practice in international journals.

Reference in manuscript: R1C2.1. The revision can be seen in the following page, Page No. 2

Reviewer's comments: Avoid long phrases; keywords should be words or short phrases, not descriptive sentences.

Authors response: R1C2.2. We appreciate the suggestion. The changes have been addressed.

Changes made: R1C2.2. We have revised the keywords accordingly. Overly long phrases have been replaced with more concise words.

Reference in manuscript: R1C2.2. The revision can be seen in the following page, Page No. 2

Reviewer's comments: Prioritize frequently used terms in indexing (for easy searchability in databases like Scopus).

Authors response: R1C2.3. Thank you for this helpful recommendation.

Changes made: R1C2.3. We have replaced the terms with frequently used and relevant terms to improve indexing and searchability in databases like Scopus.

Reference in manuscript: R1C2.3. The revision can be seen in the following page, Page No. 2

Introduction Revision Notes

Reviewer's comments: Too long and repetitive.

Authors response: R1C3.1. Thank you for the feedback.

Changes made: R1C3.1. The introduction has been streamlined by removing redundancies and keeping it more focused, shortened, and organized for better clarity.

Reference in manuscript: R1C3.1. The revision can be seen in the following pages, Page No. 3–5

Reviewer's comments: Lack of systematic structure.

Authors response: R1C3.2. We appreciate this guidance as it improves the clarity of the study.

Changes made: R1C3.2. The introduction is now reorganized to follow a clear, logical flow that guides readers effectively.

Reference in manuscript: R1C3.2. The revision can be seen in the following pages, Page No. 3–5

Reviewer's comments: Too much theory at the beginning.

Authors response: R1C3.3. We appreciate this suggestion.

Changes made: R1C3.3. We have reduced the amount of theory.

Reference in manuscript: R1C3.3. The revision can be seen in the following pages, Page No. 3–5

Reviewer's comments: Insufficiently integrated references.

Authors response: R1C3.4. We acknowledge this concern and are grateful for the observation.

Changes made: R1C3.4. Thank you for highlighting this. We have integrated references more effectively to support arguments.

Reference in manuscript: R1C3.4. The revision can be seen in the following page, Page No. 4

Reviewer's comments: Absence of explicit statement of research gaps.

Authors response: R1C3.5. Thank you for this important note.

Changes made: R1C3.5. We have added a clear statement of the research gaps to justify the relevance and contribution of the study.

Reference in manuscript: R1C3.5. The revision can be seen in the following pages, Page No. 3–5

Method Revision Notes

Reviewer's comments: Poorly organized structure. The methodology mixes method descriptions, pre-test results, and reliability analyses, which should be in the results section, not the methodology. There are overly long narrative sections (e.g., explaining students’ writing challenges) that would be more appropriate in the discussion section.

Authors response: R1C4.1. Thank you for the keen observation.

Changes made: R1C4.1. We have restructured the methodology by separating methodological details from results and moving contextual narratives to the discussion section for better clarity.

Reference in manuscript: R1C4.1. The revision can be seen in the following page, Page No. 7

Reviewer's comments: Too much operational detail. Mentioning specific intervention dates (e.g., “July 17, 2023 to September 9, 2023”) is unnecessary in the methodology unless there is a specific reason. The weekly phases are too detailed; they should be summarized as the main phases of the intervention.

Authors response: R1C4.2. We appreciate the suggestion.

Changes made: R1C4.2. We appreciate the feedback and we have removed unnecessary specifics as pointed out and condensed weekly details into broader intervention phases to keep the methodology concise.

Reference in manuscript: R1C4.2. The revision can be seen in the following page, Page No. 15

Reviewer's comments: Unclear subsections. The methodology should be divided into clear sections: Research Design, Participants, Instruments, Procedures, Data Analysis. Currently, the methodology does not follow this structure, making it difficult to follow.

Authors response: R1C4.3. Thank you for pointing this out.

Changes made: R1C4.3. The revised methodology now follows a clear structure.

Reference in manuscript: R1C4.3. The revision can be seen in the following pages, Page No. 12–23

Reviewer's comments: Lack of focus on methodological justification. There is no explanation of why mixed methods were chosen and how this method enhances the validity of the findings. There is no justification for selecting the sample size other than "average standard deviation", which needs to be linked to the research justification.

Authors response: R1C4.4. Thank you for your helpful suggestion for the enhancement of the quality of the paper.

Changes made: R1C4.4. Thank you for pointing this out. We have provided a clear rationale for adopting mixed methods and justified the sample size in relation to research objectives and the validity of findings.

Reference in manuscript: R1C4.4. The revision can be seen in the following pages, Page No. 12–13

Limitations and Implications Revision Notes

Reviewer's comments: No clear statement of limitations.

The section begins with the sentence "This study acknowledges several limitations," but then jumps to the conclusion without fully addressing the limitations of the study.

Authors response: R1C5.1. Thank you for highlighting this. It is highly appreciated.

Changes made: R1C5.1. We have revised the section to explicitly state the study's limitations before moving to the conclusion.

Reference in manuscript: R1C5.1. The revision can be seen in the following pages, Page No. 26–27

Reviewer's comments: Mixed sections. The text includes conclusions and a rehash of the results (e.g., "This study highlights the effectiveness of..."), when it should only outline the limitations of the study and implications for future research or educational practice.

Authors response: R1C5.2. We appreciate the suggestion.

Changes made: R1C5.2. We have separated limitations from conclusions and results to ensure each section serves its proper purpose.

Reference in manuscript: R1C5.2. The revision can be seen in the following page, Page No. 27

Reviewer's comments: Lack of focus on implications. The implications are still a general narrative about the effectiveness of PROMPTa, rather than clear recommendations for educators, policymakers, or future research.

Authors response: R1C5.3. Thank you for pointing out this need for clarification.

Changes made: R1C5.3. The implications have been revised to provide clear, actionable recommendations for educators, policymakers, and future researchers.

Reference in manuscript: R1C5.3. The revision can be seen in the following page, Page No. 28

Reviewer's comments: Lack of conciseness. There is a lot of repetition that should be condensed.

Authors response: R1C5.4. Thank you. It is acknowledged.

Changes made: R1C5.4. We have revised the section to make it more concise by removing unnecessary repetition and keeping the focus sharp.

Reference in manuscript: R1C5.4. The revision can be seen in the following pages, Page No. 26–27

Conclusion Revision Notes

Reviewer's comments: Too long and repetitive. Many concepts are repeated, for example, the effectiveness of PROMPTa is mentioned repeatedly.

Authors response: R1C6.1. Thank you for the feedback.

Changes made: R1C6.1. We have shortened the conclusion by removing repetitive points, especially the repeated mention of PROMPTa’s effectiveness, to keep it more focused.

Reference in manuscript: R1C6.1. The revision can be seen in the following page, Page No. 29

Reviewer's comments: Including new information that should be in "Limitations and Implications" The section on "infrastructure limitations, accessibility, and digital literacy" is more appropriately placed in the limitations/implications section, not the conclusion.

Authors response: R1C6.2. We appreciate the suggestion.

Changes made: R1C6.2. I have moved the discussion on infrastructure, accessibility, and digital literacy to the limitations/implications section, where it fits better.

Reference in manuscript: R1C6.2. The revision can be seen in the following page, Page No. 29

Reviewer's comments: Lack of focus. The conclusion should reiterate the main findings, research contributions, and brief recommendations, not rehash the methodology or lengthy discussion.

Authors response: R1C6.3. Thank you for the valuable insight.

Changes made: R1C6.3. We have revised the conclusion to highlight only the main findings, key contributions, and brief recommendations, avoiding unnecessary methodological or discussion details.

Reference in manuscript: R1C6.3. The revision can be seen in the following pages, Page No. 29–30

Reviewer's comments: Lack of academic punch line There is no clear concluding sentence about the scientific contributions and future research directions.

Authors response: R1C6.4. Thank you for the suggestion.

Changes made: R1C6.4. The revised version has strengthened the conclusion by adding a clear final sentence that highlights the study’s contributions and points toward future research directions.

Reference in manuscript: R1C6.4. The revision can be seen in the following page, Page No. 30

Reviewer 2:

Reviewer’s comments: Theoretical framework is now well-integrated, combining SCT, Constructivism, and Constructionism into a cohesive model.

Authors response: R2C1. We sincerely appreciate your suggestions that helped to strengthen our paper.

Changes made: R2C1. (NIL)

Reference in manuscript: R2C1. (NIL)

Reviewer’s comments: The methodology is clearly explained, including intervention phases and the role of PROMPTa.

Authors response: R2C2. Thank you for this acknowledgement.

Changes made: R2C2. (NIL)

Reference in manuscript: R2C2. (NIL)

Reviewer’s comments: Quantitative results are robust, and qualitative analysis provides rich contextual insights.

Authors response: R2C3. We thank you for acknowledging the contextual grounding of the study.

Changes made: R2C3. (NIL)

Reference in manuscript: R2C3. (NIL)

Reviewer’s comments: Practical relevance is high, especially for low-resource educational settings.

Authors response: R2C4. Your keen observation is deeply appreciated.

Changes made: R2C4. (NIL)

Reference in manuscript: R2C4. (NIL)

Reviewer’s comments: Inclusion of anonymized feedback samples offers transparency into the assessment process.

Authors response: R2C5. Thank you for your detailed observations.

Changes made: R2C5. (NIL)

Reference in manuscript: R2C5. (NIL)

Suggestions:

Reviewer’s comments: Consider explicitly reporting effect sizes for key statistical comparisons to complement p-values.

Authors response: R2S1. Your comments reflect both care and expertise.

Changes made: R2S1. As per the suggestion, we have reported the effect sizes alongside p-values to provide a fuller understanding of the statistical comparisons.

Reference in manuscript: R2S1. The revision can be seen in the following page, Page No. 21

Reviewer’s comments: In the Discussion, strengthen the explicit connection between findings and how they advance or challenge prior literature.

Authors response: R2S2. Thank you for pointing this out. It points the work in the right direction for further refinement.

Changes made: R2S2. As per the suggestions, we have strengthened the discussion by explicitly linking the findings to how they support, extend, or challenge previous studies.

Reference in manuscript: R2S2. The revision can be seen in the following page, Page No. 26

Reviewer’s comments: Briefly address scalability and feasibility of PROMPTa in varied institutional contexts, especially where ICT infrastructure is limited.

Authors response: R2S3. Thank you for this point.

Changes made: R2S3. We have briefly addressed the scalability of PROMPTa and its feasibility in contexts with limited ICT infrastructure.

Reference in manuscript: R2S3. The revision can be seen in the following page, Page No. 28

Reviewer’s comments: Split a few longer paragraphs in the Introduction and Literature Review for improved readability.

Authors response: R2S4. Thank you for highlighting this.

Changes made: R2S4. We have split longer paragraphs in these sections to enhance readability and flow.

Reference in manuscript: R2S4. The revision is evident in the mentioned sections.

---

## [Decision Letter · Decision Letter 2]

15 Sep 2025

Dear Dr. Nittala, 

Thank you for submitting your manuscript to PLOS ONE. After careful consideration, we feel that it has merit but does not fully meet PLOS ONE’s publication criteria as it currently stands. Therefore, we invite you to submit a revised version of the manuscript that addresses the points raised during the review process.

We look forward to receiving your revised manuscript.

Kind regards,

Wei Lun Wong

Academic Editor

PLOS ONE

Journal Requirements:

Reviewers' comments:

Reviewer's Responses to Questions

**Comments to the Author**

Reviewer #1: (No Response)

Reviewer #2: All comments have been addressed

2. Is the manuscript technically sound, and do the data support the conclusions?

Reviewer #1: Yes

Reviewer #2: Yes

3. Has the statistical analysis been performed appropriately and rigorously?

Reviewer #1: I Don't Know

Reviewer #2: Yes

4. Have the authors made all data underlying the findings in their manuscript fully available?

Reviewer #1: Yes

Reviewer #2: Yes

5. Is the manuscript presented in an intelligible fashion and written in standard English?

Reviewer #1: Yes

Reviewer #2: Yes

Reviewer #1: Abstract Revision Notes

1. IMRAD Structure

International journal abstracts are generally concise and follow a clear flow of Introduction–Method–Result–Conclusion. Your abstract includes all of these sections, but the transitions are still long and rambling. They need to be condensed and emphasized.

2. Clarity of Purpose

The purpose statement is scattered throughout the abstract. It should be emphasized in one clear sentence: “This study aims to evaluate…”.

3. Methodology

The information is present (mixed-methods, number of participants, randomization), but it is too long. International abstracts are usually concise, simply mentioning the design, participants, and instruments briefly.

4. Main Results

The results are presented, but the phrase “significant improvement” should be reinforced with numbers (e.g., post-test mean score, effect size, or p-value). If numbers are not present, at least use more academic language.

5. Contribution/Implications

Present, but somewhat repetitive. Could be condensed to focus on novelty and practical implications.

6. Language

Needs to be more academic, avoiding general terms like "adds to existing literature" → replace with "extends existing literature on technology-assisted assessment in writing pedagogy."

Use active but formal sentences, and avoid repeating "technology-based assessment" more than once.

Keyword Revision Notes

1. Quantity: International journals typically have 4–6 keywords, no more. Your keywords are too many (7).

2. Consistency with the Abstract: Keywords should be consistent with the terms appearing in the abstract.

Avoid excessive use of synonyms. For example, Peer Feedback, Mentor Feedback, and Expert Feedback can be condensed into one general term, such as Peer and Expert Feedback.

3. Global Readability: Because the target is international, keywords should be in English and should not use abbreviations without explanation (PROMPTa is fine, but TIK is better changed to ICT).

Introduction Revision Notes

1. Structure and Flow

The introduction is too long and contains a mix of general background, literature review, and local context.

It should be organized in a clear flow:

Paragraph 1: The importance of writing in a global/academic context (general gap).

Paragraph 2: Challenges in writing assessment, especially in the digital/AI era (literature gap).

Paragraph 3: Local context (Nagaland, multilingualism, education system, inequality).

Paragraph 4: Research gap + research objective (this study aims to…).

Currently, the Research Background section can be condensed into paragraph 3 (no need for a separate subheading, as introductions in international journals typically do not use headings).

2. Clarity of the Research Gap

The introduction is still descriptive (describing the general situation and challenges of writing).

There needs to be explicit sentences that indicate:

(1) What has been researched by previous studies,

(2) What has not been researched (gap),

(3) How this research fills the gap.

Example:

“While prior studies have examined writing pedagogy in multilingual settings [15–17], limited research has explored technology-assisted formative assessment as a means to improve writing proficiency in the context of Nagaland. This study addresses this gap…”

3. Redundancy and Excessive Detail

Some sentences are too detailed for an introduction (e.g., about Nagamese, teachers dictating notes, one-way traffic). These details should be condensed or moved to the discussion section.

For example:

“In Nagaland, the dominance of teacher-centered practices and reliance on rote memorization have limited students’ opportunities to develop writing as a process of critical thinking and creativity [17–20].”

4. International Academic Language

Currently, the language is still narrative-descriptive and needs to be condensed with an analytical style.

Examples of less academic phrases: “The current system has failed to empower learners” → should be: “The current system remains predominantly teacher-centered, limiting learner autonomy and critical engagement.”

5. Relevance to the Study

The section on AI and academic authenticity is interesting, but it doesn't directly connect to the research focus (PROMPTa as a solution). There needs to be a stronger transition so readers see that this study isn't just describing a problem, but also offering a solution.

6. Research Objective Statement

It should be explicit at the end of the introduction. Currently, the objective appears vague.

Add one final sentence, for example:

“Therefore, this study aims to investigate the effectiveness of PROMPTa, a multifaceted ICT-based formative assessment tool, in enhancing writing skills among first-year students in Nagaland.”

Method Revision Notes

1. Clarity and Consistency of Research Design

The text uses the terms mixed methods, convergent parallel design, and quasi-experimental design simultaneously. This may confuse international readers.

Revision: The relationship between the two needs to be clarified. For example, "This study used a mixed methods design with a quantitative component in the form of a quasi-experimental design and a qualitative component in the form of interviews, questionnaires, and observations."

2. Justification for Method Selection

The methods are currently described, but the rationale for selecting convergent mixed methods and quasi-experimental design is not yet strong.

Revision: Add methodological references (e.g., Creswell & Plano Clark, 2018; Johnson & Onwuegbuzie, 2004) to explain why this combination of methods is appropriate for the research question.

3. Participant Description

Participant data is quite detailed, but lacks international standards.

Revision: Add:

Participant inclusion and exclusion criteria.

Randomization process (e.g., simple random sampling or stratified).

Gender ratio (how many men, how many women).

Justify sample size using power analysis (e.g., G*Power).

4. Research Instrument

The instrument is describe1d at length, but remains narrative.

Revision: Create a summary table of the instrument that includes: instrument type, purpose, data format, sample items/activities, and validation sources.

Instrument validity (content validity) is only mentioned based on expert input; it should be referred to as a formal method such as the Content Validity Index (CVI).

5. Reliability and Validity

Reliability is referred to as Pearson correlation, although for rubric assessment, Cronbach's Alpha or Inter-rater Reliability (Cohen's Kappa/ICC) should be used.

Revision: Replace or add appropriate reliability techniques.

6. PROMPTa Implementation

The PROMPTa description is too long; it would be better included in the Instrument section or Appendix.

Revision: Condense the description in the Methodology and move technical details to the Appendix so readers don't lose focus on the method's structure.

7. Research Stages

The week-by-week explanation is very narrative.

Revision: Create a table or research flowchart to clarify the 8-week intervention timeline.

8. Data Analysis

The quantitative analysis is clear (Levene's test, t-test), but the effect size (Cohen's d, η²) is not mentioned. This is despite international journals requiring the reporting of effect sizes.

The qualitative analysis is only called "thematic analysis," but the detailed procedures are not explained (e.g., Braun & Clarke, 2006: coding, theme development, triangulation).

Revision: Add detailed qualitative analysis procedures and the software used (e.g., NVivo/Atlas.ti).

9. Research Ethics

Ethics approval is in place, but it would be better to add the ethics clearance number and a reference to international ethical standards (e.g., the Declaration of Helsinki or APA Ethical Principles).

10. International Academic Language

Currently, the text still seems lengthy and descriptive. For international journals, the methodology needs to be concise, systematic, and evidence-based.

Revision: Use consistent subheadings (Research Design, Participants, Instruments, Procedures, Data Analysis, Ethical Considerations).

Discussion of Results Revision Notes

1. Too Descriptive, Not Analytical

Many sections simply repeat figures from tables/figures (M, SD, p-value, correlation), whereas in international journals, these should be confined to the results section, not the discussion.

The discussion should emphasize what the figures mean, not simply repeat them.

Revise → summarize the reported figures and focus on their meaning:

“The strong consistency of intra- and inter-rater reliability confirms the credibility of the evaluation instrument, strengthening the reliability of the experimental results in the context of ICT-based assessment.”

2. Lack of Linking to Previous Literature

The discussion should compare the findings with previous research (e.g., whether the results align with/contradict other studies).

Currently, there are only internal claims without explicit references.

Revise → insert citations:

“These findings are consistent with study X (2020) which showed that the use of ICT-based platforms can improve the quality of feedback and students' writing motivation.”

3. Language Remains Local, Lacking Globality

Discussions tend to focus solely on the Nagaland context.

International journals expect generalizations or implications for global contexts (e.g., other developing countries with digital limitations).

Revise → withdraw generalizations:

“Although the study was conducted in Nagaland, the challenges faced—limited digital access, low motivation to write, and lack of meaningful feedback—are also relevant in many developing regions in Asia and Africa.”

4. Underemphasize Practical Significance

Cohen’s d = 1.25 is provided, but it doesn’t explain what it means in educational practice.

International readers want to know: is this a large effect size worth considering in policy?

Revise → make a practical interpretation:

“The large effect size (Cohen’s d = 1.25) confirms that PROMPTa integration is not only statistically significant but also has a substantial practical impact on writing skill development.”

5. Qualitative Analysis Lacks Integration with Quantitative

Qualitative results (themes, participant quotes) are still presented separately from quantitative results.

International journal standards typically require triangulation → demonstrating how qualitative data supports or enriches quantitative data.

Revise → integrate:

“The improvement in post-test scores in the experimental group is reinforced by qualitative findings: participants emphasized that PROMPTa's feedback structure reduced anxiety and improved writing clarity, consistent with the significant shift in quantitative scores.”

6. Lack of Critical Discussion

There is no discussion of why there was variation in scores (e.g., the experimental group's SD was higher → indicating a non-uniform effect).

There is no critical reflection on the limitations of the results (e.g., could the effect have diminished without mentor support?).

Revise → add critical reflection:

“Although scores improved significantly, the higher variability in the experimental group suggests that PROMPTa's impact was uneven. Factors such as differences in digital readiness and individual motivation may have influenced the intervention's effectiveness.”

Conclusion Revision Notes

1. More concise and focused – The current conclusion is too long and mixed with narrative “discussions” (e.g., about adaptation in low-connectivity areas). International journals typically require concise conclusions, not repeating details already discussed.

2. More systematic structure – The conclusion should consist of:

Summary of the main findings.

Theoretical and practical implications.

Limitations of the study (briefly, if not a separate section).

Directions for future research.

3. Objective academic language – Phrases such as “the evidence that alternative assessment is practically effective” are too normative. It is better to use evidence-based language, such as “the findings demonstrate the effectiveness of alternative assessment in…”.

4. Avoid repetition – Excessive repetition (e.g., “improved writing efficiency” appears more than once). Only mention it once at the beginning.

5. Relationship to the literature – By international standards, the conclusion should emphasize how the study's results fill a gap in previous research or reinforce the findings of other researchers.

6. Explicit limitations & recommendations – Currently not mentioned, although almost all international journals require acknowledgment of study limitations and suggestions for further research.

Implications Revision Notes

1. Language remains normative and repetitive

Words like "encouraging" and "can have a positive impact" are too general and lack academic relevance.

Revise → Use language based on empirical findings, for example: "The findings demonstrate the potential effectiveness of implementing ICT-based alternative assessments in the long term to support the development of students' writing skills."

2. Lack of explicit separation of practical and theoretical implications

The implications are written for researchers, educators, and policymakers, but do not clearly differentiate between practical and theory-based aspects.

Revise → Use subheadings "Practical Implications" and "Policy Implications" for greater systematic presentation.

3. Lack of clarity in connecting the findings to the global context

The implications are still focused on Nagaland/India. For international journals, it would be better to draw a broader context: how these results are relevant in digitally challenged regions globally (Southeast Asia, Africa, Latin America).

4. Lack of linking to previous literature

International journals typically require that implications be based not only on their own findings but also on previous studies.

Theoretical Contributions Revision Notes

1. Lack of specificity regarding theoretical gaps

Currently, the paper only mentions Social Cognitive Theory (SCT) and Constructivism, but does not explain the gaps in previous research and how this study addresses them.

Revise → add a brief explanation: "Previous studies on the application of SCT and constructivism to writing instruction have been limited to face-to-face/traditional contexts. This study expands the application of these theories to the ICT-based digital realm, specifically in alternative assessment."

2. Contribution is still descriptive, not conceptual.

It should be emphasized that this research produces a new framework, conceptual model, or operationalization of theory in a technological context.

Revise → explicitly state the conceptual contribution (e.g., "This contribution extends SCT by emphasizing the digital dimension of peer feedback").

3. The language is too long and narrative.

It should be more concise, systematic, and use an academic structure:

Theory used.

Existing gap.

How this research fills the gap.

New model/contribution.

4. Not enough emphasis on global relevance.

While it is simply called “21st century,” it should emphasize that this theoretical contribution is relevant across global educational contexts, not just in Nagaland.

Reviewer #2: The authors have substantially improved the manuscript in response to previous review rounds. The abstract is now concise, clearly states the research gap, objectives, and implications. The introduction is streamlined and follows a logical flow, with reduced redundancy and a clear articulation of the research gap.

The methodology section has been reorganized into well-defined subsections (Research Design, Participants, Instruments, Procedures, Data Analysis), improving readability and reproducibility. The justification for using a mixed-methods approach and sample size has been added, strengthening methodological rigor.

The results are robust and complemented by effect sizes, enhancing interpretability. The discussion now effectively links findings to prior literature and emphasizes theoretical and practical implications. Limitations and implications are clearly separated, with actionable recommendations for educators and policymakers.

The conclusion is more concise, avoids repetition, and provides a strong final sentence that highlights contributions and future research directions.

Overall, this revised manuscript is now of publishable quality and meets the PLOS ONE criteria for scientific and methodological rigor.

**Do you want your identity to be public for this peer review?** For information about this choice, including consent withdrawal, please see our Privacy Policy

Reviewer #1: No

Reviewer #2: **Yes: ** Alexander Adrian Saragi

---

## [Author Response · Author response to Decision Letter 3]

24 Sep 2025

Response to Reviewers

Dear Editors and Reviewers,

We are thankful for the thorough and thoughtful responses to the invaluable feedback the Editor and Reviewers have provided. Each comment has been carefully considered, and we have taken diligent measures to address every query raised. Our commitment to improving the quality of this manuscript is evident in the revisions we have made.

On behalf of all the authors, I express my sincere gratitude to the Reviewers and the esteemed Editor for their constructive input and valuable suggestions. Their guidance has undoubtedly played a significant role in refining this piece of work. We are genuinely appreciative of this opportunity to enhance our research, and we eagerly look forward to sharing the enhanced and polished version with the esteemed readership of the journal in the near future.

All the changes have been uploaded in the file named Response to the Reviewers in the submission portal.

Sincerely,

Noel.

---

## [Decision Letter · Decision Letter 3]

23 Oct 2025

Dear Dr. Nittala,

Thank you for submitting your manuscript to PLOS ONE. After careful consideration, we feel that it has merit but does not fully meet PLOS ONE’s publication criteria as it currently stands. Therefore, we invite you to submit a revised version of the manuscript that addresses the points raised during the review process.

We look forward to receiving your revised manuscript.

Kind regards,

Wei Lun Wong

Academic Editor

PLOS ONE

Journal Requirements:

Reviewers' comments:

Reviewer's Responses to Questions

**Comments to the Author**

Reviewer #2: All comments have been addressed

2. Is the manuscript technically sound, and do the data support the conclusions?

Reviewer #2: Yes

3. Has the statistical analysis been performed appropriately and rigorously?

Reviewer #2: Yes

4. Have the authors made all data underlying the findings in their manuscript fully available?

Reviewer #2: Yes

5. Is the manuscript presented in an intelligible fashion and written in standard English?

Reviewer #2: Yes

Reviewer #2: The manuscript demonstrates a well-conceived and executed study on ICT-integrated multifaceted assessment for writing skills. The integration of peer, mentor, and expert feedback within the PROMPTa framework is a timely and valuable contribution to technology-enhanced learning literature. The study effectively combines quantitative and qualitative analyses, presenting compelling evidence that ICT-mediated assessment improves student writing proficiency and engagement.

Strengths:

- Strong theoretical grounding in Social Cognitive Theory, Constructivism, and Constructionism.

- Methodologically rigorous mixed-methods design.

- Clear articulation of intervention procedures and validation of instruments.

- Insightful discussion linking data to policy contexts such as NEP 2020.

Suggestions for minor improvement:

- Include more details on how PROMPTa ensures inter-rater fairness and consistency in real classroom settings.

- Clarify whether the tool’s usability and accessibility were measured quantitatively (e.g., user satisfaction scales).

- A brief reflection on scalability or adaptability to other educational contexts could strengthen the implications section.

Overall, this manuscript meets PLOS ONE’s publication criteria. I recommend acceptance after minor language and clarification revisions.

**Do you want your identity to be public for this peer review?** For information about this choice, including consent withdrawal, please see our Privacy Policy

Reviewer #2: **Yes: ** Alexander Adrian Saragi

---

## [Author Response · Author response to Decision Letter 4]

24 Oct 2025

Dear Reviewer,

We thank you for your constructive feedback on our manuscript. We greatly appreciate the time and the effort taken to highlight areas requiring revision. We have carefully considered each of your suggestion and made revision accordingly. Thank you for your invaluable inputs in making this manuscript ready for publication as per the International standards.

Comments

1. Include more details on how PROMPTa ensures inter-rater fairness and consistency in real classroom setting.

Authors’ response

We thank the reviewer for this insightful suggestion. The revised section now elaborates on how PROMPTa ensures inter-rater fairness and consistency in real classroom contexts.

Changes made

To ensure fairness and consistency in scoring, PROMPTa followed several accuracy checks. In this context, there were three levels of assessors: peers, mentors, and experts. Peers received orientation and practice using sample writings and clear explanations of the rubrics created by the researcher. The mentor, who was also the researcher, applied the same validated rubrics when providing scoring and feedback. Experts, who had validated the rubrics, also evaluated the final drafts to ensure consistency across all stages. While scoring PROMPTa offered a structured digital platform where all assessors could refer to the same rubrics. Student identities remained anonymous to prevent bias, and the platform encouraged specific, evidence-based feedback instead of subjective opinions.

Reference in Manuscript

(Page Numbers)

The revision can be seen in the following page, Page No. 13

2. Clarify whether the tool’s usability and accessibility were measured quantitatively (e.g., user satisfaction scales)

Authors’ response

We appreciate the reviewer’s helpful observation. The revision clarifies that while standardised user-satisfaction scales were not employed, usability and accessibility were examined qualitatively through open–ended questionnaires and interviews.

Changes made

Usability and accessibility were examined qualitatively through open-ended questionnaires and interviews focusing on navigation, ease of use, and accessibility of submission, feedback, and anonymity features. Though no standardized user-satisfaction scales were employed, post-intervention responses indicated strong usability: 82% found the prompts engaging and 88% appreciated the platform’s accessibility. These, supported by qualitative feedback show that PROMPTa was both user-friendly and accessible.

Reference in Manuscript

(Page Numbers)

R1C1.1. The revision can be seen in the following page, Page No. 13

3. A brief reflection on scalability or adaptability to other educational contexts could strengthen the implications section.

Authors’ response

We thank the reviewer for this valuable recommendation. A concluding point has been added to reflect PROMPTa’s scalability and adaptability.

Changes made

Scalability and Adaptability: PROMPTa demonstrates strong potential for adaptation beyond its original setting. Its rubric based, flexible design allows customization across subjects, grade levels, and institution types.

Reference in Manuscript

(Page Numbers)

The revision can be seen in the following page, Page No. 27

---

## [Decision Letter · Decision Letter 4]

6 Nov 2025

PROMPTa - A Multifaceted Assessment Approach to Develop Writing Skills

PONE-D-25-23770R4

Dear Dr. Noel, 

We’re pleased to inform you that your manuscript has been judged scientifically suitable for publication and will be formally accepted for publication once it meets all outstanding technical requirements.

Kind regards,

Wei Lun Wong

Academic Editor

PLOS ONE

Additional Editor Comments (optional):

Reviewers' comments:

Reviewer's Responses to Questions

**Comments to the Author**

Reviewer #2: All comments have been addressed

2. Is the manuscript technically sound, and do the data support the conclusions?

Reviewer #2: Yes

3. Has the statistical analysis been performed appropriately and rigorously?

Reviewer #2: Yes

4. Have the authors made all data underlying the findings in their manuscript fully available?

Reviewer #2: Yes

5. Is the manuscript presented in an intelligible fashion and written in standard English?

Reviewer #2: Yes

Reviewer #2: The authors have adequately addressed the comments raised in the previous review round. The revisions improve the clarity of the methodology, the explanation of PROMPTa, and the alignment between objectives, data analysis, and conclusions.

The statistical procedures (pre-test/post-test comparison, reliability estimates, Levene’s test, t-tests) are now clearly presented and appropriately interpreted. The addition of both intra-rater and inter-rater reliability strengthens the validity of the findings.

The qualitative analysis section has been clarified and now logically complements the quantitative results. The thematic analysis is well-structured, and its connection to the research questions is clearer than in the earlier draft.

Minor language issues remain (e.g., occasional long sentences and typographical spacing), but they do not affect readability or scientific interpretation. These can be corrected during final proofreading.

Overall, the manuscript is now suitable for publication.

**Do you want your identity to be public for this peer review?** For information about this choice, including consent withdrawal, please see our Privacy Policy

Reviewer #2: **Yes: ** Alexander Adrian Saragi

---

## [Editor Report · Acceptance letter]

PONE-D-25-23770R4

PLOS ONE

Dear Dr. Nittala,

I'm pleased to inform you that your manuscript has been deemed suitable for publication in PLOS ONE. Congratulations! Your manuscript is now being handed over to our production team.

Kind regards,

on behalf of

Dr. Wei Lun Wong

Academic Editor

PLOS ONE